# The Impact of Hurricane Disturbances on a Tropical Forest: Implementing a Palm Plant Functional Type and Hurricane Disturbance Module in ED2-HuDi V1.0

Jiaying Zhang[1], Rafael L. Bras[1], Marcos Longo[2,3], Tamara Heartsill Scalley[4]

[1]School of Civil and Environmental Engineering, Georgia Institute of Technology, Atlanta, GA, United States
[2]Jet Propulsion Laboratory, California Institute of Technology, Pasadena, CA, United States
[3]Climate and Ecosystem Sciences Division, Lawrence Berkeley National Laboratory, Berkeley, CA, United States
[4]USDA Forest Service, International Institute of Tropical Forestry, Río Piedras, PR, United States.

*Correspondence to*: Jiaying Zhang (jiaying.zhang@gatech.edu); Rafael L. Bras (rlbras@gatech.edu)

**Abstract**

Hurricanes commonly disturb and damage tropical forests. Hurricane frequency and intensity are predicted to change under the changing climate. The short-term impacts of hurricane disturbances to tropical forests have been widely studied, but the long-term impacts are rarely investigated. Modeling is critical to investigate the potential response of forests to future disturbances, particularly if the nature of the disturbances is changing with climate. Unfortunately, existing models of forest dynamics are not presently able to account for hurricane disturbances. Therefore, we implement the Hurricane Disturbance in the Ecosystem Demography model (ED2) (ED2-HuDi). The hurricane disturbance includes hurricane-induced immediate mortality and subsequent recovery modules. The parameterizations are based on observations at the Bisley Experimental Watersheds (BEW) in the Luquillo Experimental Forest in Puerto Rico. We add one new plant functional type (PFT) to the model—Palm, as palms cannot be categorized into one of the current existing PFTs and are known to be an abundant component of tropical forests worldwide. The model is calibrated with observations at BEW using the generalized likelihood uncertainty estimates (GLUE) approach. The optimal simulation obtained from GLUE has a mean relative error of -21%, -12%, and -15% for stem density, basal area, and aboveground biomass, respectively. The optimal simulation also agrees well with the observation in terms of PFT composition (+1%, -8%, -2%, and +9% differences in the percentages of Early, Mid, Late, and Palm PFTs, respectively) and size structure of the forest (+0.8% differences in the percentage of large stems). Lastly, using the optimal parameter set, we study the impact of forest initial condition on the recovery of the forest from a single hurricane disturbance. The results indicate that, compared to a no-hurricane scenario, a single hurricane disturbance has little impact on forest structure (+1% change in the percentage of large stems) and composition (< 1% change in the percentage of each of the four PFTs) but leads to 5% higher aboveground biomass after 80 years of succession. The assumption of a less severe hurricane disturbance leads to a 4% increase in aboveground biomass.

## 1    Introduction

Hurricanes are an important disturbance agent in tropical forests. They damage individual trees and reduce aboveground biomass (Zimmerman et al. 1994; Uriarte et al. 2019; Rutledge et al. 2021; Leitold et al. 2021). For example, hurricane Hugo in 1989 uprooted and snapped 20% of the trees at El Verde in the Luquillo Experimental Forest (LEF), Puerto Rico (Walker 1991; Walker et al. 1992; Zimmerman et al. 1994) and reduced the aboveground biomass by 50% at Bisley in the LEF (Scatena et al. 1993; Heartsill Scalley et al. 2010). Hurricane Katrina in 2005 damaged about 320 million large trees on U.S. Gulf Coast forests, and the damaged trees are equivalent to 50-140% of the net annual U.S. carbon sink (Chambers et al. 2007). In the long term, the recovery from those damages will alter forest species composition and structure (Royo et al. 2011; Heartsill Scalley 2017).

Hurricane-induced mortality varies with many factors, including hurricane severity (Parker et al. 2018), environmental conditions (Uriarte et al. 2019; Hall et al. 2020), forest exposure to hurricane winds (Boose et al. 1994; Boose et al. 2004), forest structure (Zhang et al. 2022b), and traits and size of individual trees (Curran et al. 2008; Lewis and Bannar-Martin 2011). Trees with a larger diameter have been found to be more resistant to wind forces but more likely to suffer broken branches (Lewis and Bannar-Martin 2011). Species with higher wood density tend to

suffer less from hurricane disturbances (Zimmerman et al. 1994; Curran et al. 2008). Hurricanes with heavier rainfall
and stronger wind generally lead to higher mortality (Uriarte et al. 2019; Hall et al. 2020), and forests that are more
exposed to strong winds tend to have higher mortality (Uriarte et al. 2019). However, forests with a more wind-
resistant structure and composition experience lower mortality even during a stronger hurricane event or a higher
exposure (Zhang et al. 2022b).

The recovery from hurricanes also depends on many factors, such as the disturbance severity (Walker 1991;
Everham and Brokaw 1996; Cole et al. 2014; Heartsill Scalley 2017) and traits of individual species (Curran et al.
2008; Lewis and Bannar-Martin 2011). Species with lower wood density have shorter times to resprout (Paz et al.
2018), higher growth rate (King et al. 2006), and shorter biomass recovery times (Curran et al. 2008). The number of
resprouts of some species further varies with time since disturbance (Brokaw 1998). Less severe disturbances lead to
a faster recovery and a higher recovery of stem density and aboveground biomass compared to the level observed
prior to the disturbance (Wang and Eltahir 2000; Parker et al. 2018). For example, observations on a tropical forest
canopy in western Mexico after two hurricanes—category 2 Jova and category 4 Patricia—showed that hurricane Jova
destroyed 11% of the aboveground biomass while hurricane Patricia destroyed 23%; the recovery was more rapid
after the less intense hurricane Jova (Parker et al. 2018). Although the immediate mortality and subsequent recovery
of tropical forest from hurricane disturbances have been thoroughly studied via observations, the long-term effects of
consecutive hurricane disturbances on tropical forests have rarely been studied. Models that can simulate the
immediate mortality and subsequent recovery of an ecosystem can play a role in understanding potential mechanisms
driving the mortality and recovery of the ecosystems and studying the long-term effects of disturbances, particularly
if the nature of the disturbances is changing with climate. Uriarte et al. (2009) implemented hurricane disturbance in
a forest simulator and investigated the long-term dynamics of forest composition, diversity, and structure. However,
the biological and environmental processes of the forest simulator used are not dynamic and thus the model cannot
simulate the adaptation of vegetation to the changes of environment (Jorgensen 2008). Vegetation dynamics models
can account for changes in the ecosystem resulting from a changing environment (Medvigy et al. 2009; Longo et al.
2019b), and further allow us to explore scenarios via synthetic experiments and thus emulate what might happen in
forests under novel environmental conditions. For example, Feng et al, (2018) used the Ecosystem Demography model
(ED2) (Moorcroft et al. 2001) to study the impact of climate change on the forest studied in Uriarte et al. (2009).  The
ED2 model is a process-based vegetation dynamics model, it represents the size and age structure of the forest, and
thus the model can represent the observed differential impact from disturbances (such as fire, drought, insects, land
use change, and natural disturbances) across plants of different functional groups and size classes (Medvigy et al.
2012; Zhang et al. 2015; Miller et al. 2016, Trugman et al. 2016). However, the impacts of hurricane disturbances
have not been implemented in vegetation dynamics models, and thus the long-term effects on the forest of a changing
hurricane regime have not been investigated.

As mortality and recovery vary with species, the species composition of the forest is affected by hurricane
disturbances. In modeling studies, it is impractical to incorporate each and individual species (tens and hundreds). To
address variation in species diversity, there has been a strong effort in the past decades to incorporate functional
diversity in vegetation dynamics models (Moorcroft et al. 2001; Sakschewski et al. 2016; Fisher et al. 2018; Fisher
and Koven 2020). This effort acknowledges the variability in traits and trade-offs of species that exist in tropical
forests (e.g., Baraloto et al. 2010). Three plant functional types (PFT) are identified for the species in tropical forests
during a secondary succession after a disturbance; they are early, mid, and late successional PFTs (hereafter Early,
Mid, and Late PFTs), corresponding to the three successional stages during the secondary succession (Kammesheidt
2000). Specifically, Early PFT dominates the early successional stage of the recovery, it includes fast growing pioneer
species that have low wood density, establish and recruit in open gaps formed after disturbances and grow rapidly in
the high light environment. Mid PFT dominates the mid successional stage after a disturbance, and includes species
that have intermediate growth and are somewhat shade tolerant. Late PFT dominates the late successional stage and
includes species that have slow growth and are shade tolerant. Using three PFTs is also a compromise between
representing a range of life strategies while not adding too much complexity in model parameterizations (Moorcroft
et al. 2001; Medlyn et al. 2005).

One important and distinct species in tropical forests in the Caribbean islands is the palm species *Prestoea*

*montana* (Sierra palm). Many studies in the Luquillo Mountains have either excluded palms from analysis
(Zimmerman et al. 1994) or treated palms separately from other trees (Zimmerman et al. 1994; Uriarte et al. 2009), as
indeed they are monocots, not dicots like the other trees in the forest. A previous study that simulates the response of
the forests in the Luquillo Mountains to climate change using the ED2 model categorized the palm species as a Late
PFT tree (Feng et al. 2018). However, there are important differences, palms are more resistant to hurricane damage
as compared to trees (Francis and Gillespie 1993; Uriarte et al. 2019) and are more resilient to hurricane disturbances
due to their high fecundity under open canopy (Lugo and Rivera Batlle 1987; Lugo et al. 1998) and have high tolerance
to shade (Ma et al. 2015). All those characteristics separate palms from other trees and favor the survival of palms
after hurricane disturbances. We believe palms cannot be categorized into one of the existent PFT categories in the
model, and hence we define a new PFT—Palm.

In this paper, we describe the implementation of hurricane mortality and recovery modules that account for

the variation with disturbance severity, forest resistance state, PFT and diameter size of individual stems in the
Ecosystem Demography model (ED2). The model is then used to study the recovery of a tropical rainforest after
hurricane disturbances. The results indicate that a scenario with a single hurricane disturbance has little long-term
impact on forest structure and composition but enhances the aboveground biomass accumulation of a tropical
rainforest, relative to a no hurricane disturbance scenario.
**2    Methods and Materials**
**2.1    Census Observations**
Tree censuses were carried out at Bisley Experimental Watersheds (BEW) in the Luquillo Experimental Forest in
Puerto Rico starting in 1989, three months before hurricane Hugo (pre-Hugo 1989), and repeated three months after
hurricane Hugo (post-Hugo 1989), and then every five years since then (1994, 1999, 2004, 2009, 2014). The census
recorded the diameter at breast height (1.3m) (DBH) and species of each stem with DBH ≥ 2.5 cm and height (H) of
selected stems in 85 permanent forest dynamics plots in the forest. Each plot is a 10-meter diameter circle and plots
are 40 meters apart extending 13 hectares. The last census was conducted three months after hurricane Maria and
recorded auxiliary damage information of each stem. The detailed description of the study site and the census
observations can be found in Zhang et al. (2022b) and the census data between 1989 and 2014 are from Zhang et al.
(2022a) and the post-Maria census data are from Zhang et al. (2020). Following Zhang et al. (2022b), species are
categorized into four PFTs according to their successional status based on previous studies (Walker 1991; Schowalter
and Ganio 1999; Uriarte et al. 2005; Muscarella et al. 2013; Heartsill Scalley 2017; Feng et al. 2018): early, mid, late
successional tropical trees, and palms (Early, Mid, Late, and Palm PFT, respectively). The stem density, DBH growth
rate, and basal area are calculated from the census data for each PFT in each census. The aboveground biomass (AGB)
of Early, Mid, and Late PFTs are estimated from DBH using the AGB-DBH relationship from Scatena et al. (1993);
the AGB of Palm PFT is estimated from the AGB-Height relationship of *P. montana* from Scatena et al. (1993) and
the Height-DBH relationship of Palm PFT from the census observations at our study site (Section 2.2.2).
**2.2    Model Description**
The Ecosystem Demography model (ED) is a cohort-based model, and it describes the growth, reproduction, and
mortality of each cohort in each patch in a forest site. A cohort is a group of stems with the same PFT and similar
diameter size and age. A patch is an area with the same environmental condition and disturbance history. A cohort
accumulates carbon through photosynthesis, and the net accumulated carbon (i.e., gross primary productivity minus
respiration and maintenance of living tissues) will be used for growth and reproduction. When a cohort is mature,
reaching the maturity reproductive height (e.g., 18 m), the cohort will allocate a portion of carbon to reproduction
(e.g., 30% of net carbon accumulation to seeds, flowers, and fruits), and the rest of the net accumulated carbon will
be used for structural growth. Structural growth is quantified by the increase of DBH through structural biomass-DBH
allometries; stem height, leaf biomass, and crown area are then scaled given the H-DBH, leaf biomass-DBH, and
crown-DBH allometries. Each cohort will also experience mortality from multiple factors, including aging,
competition, and disturbance, which will be described in detail in Section 2.3.2.

The model simulates transient fluxes of carbon, water, and energy during short-term physiological responses

and long-term ecosystem composition and structure responses to changes in environmental conditions. The second
version of the ED model, ED2, modifies the calculations of radiation and evapotranspiration of the original ED model,
leading to a more realistic long-term response of ecosystem composition and structure to atmospheric forcing
(Medvigy et al. 2009; Longo et al. 2019b). Details of the ED and ED2 models can be found in Moorcroft et al. (2001),
Medvigy et al. (2009), and Longo et al. (2019a). Here we add a new PFT (Palm) and implement hurricane disturbance
in the ED2 model, and we name it ED2-HuDi V1.0.
**2.2.1    Adding Palm as a New PFT**
The standard ED2 model represents a variety of broadleaf trees, needleleaf trees, grasses and lianas (Albani et al.
2006; Medvidy et al. 2009; Longo et al. 2019a; di Porcia e Brugnera et al. 2019). Yet, to date, none of the existing
PFTs describe the traits of palms, even though palms are a globally abundant component of tropical forests (Muscarella
et al. 2020). We know that the palm species that occurs at our study site (*Prestoea montana*) has a low wood density
of 0.31 g cm$^{-3}$ (Swenson and Umana 2015) and it grows fast in open canopies like early successional tropical trees
(Lugo and Rivera Batlle 1987; Lugo et al. 1998) and are tolerant to shade like late successional tropical trees (Ma et
al. 2015). Hence, we assume that the physiological traits of Palm have the same probability distributions as those of
Late PFT except for wood density which is assumed the same as that of Early PFT. The allometries of Palm are
discussed separately in the next section.

### 2.2.2    Modifying the Allometric Relationship

The allometric relationships between stem height (H; m) and diameter at breast height (DBH; cm) for four tropical
PFTs (Early, Mid, Late, and Palm) come from census data at BEW in the Luquillo Experimental Forest in Puerto Rico
(Zhang et al. 2022a). The relationships take the form,

$$H = a\,DBH^{b}\,, \tag{1}$$

where $a$ and $b$ are PFT-specific scale and shape parameters (Figure 1). The diameter range for the Palm PFT is between
10 and 20 cm while that for the tree PFTs is between 2.5 and 90 cm. The scale parameter $a$ is 1.6388, 2.2054, 2.3833,
and 0.1628 for Early, Mid, Late, and Palm PFT, respectively. The shape parameter $b$ is 0.80, 0.64, 0.59, and 1.47 for
the four PFTs (Table S1). Palm has a smaller scale parameter and a significantly larger shape parameter, demonstrating
that palms are shorter than other PFTs given the same DBH. The constrained diameter range and the H-DBH allometry
of Palm make it difficult for palms to access sunlight and would normally prevent them from establishing in the ED2
model. A previous study implementing liana to the ED2 model also experienced similar issues (di Porcia e Brugnera
et al. 2019). They used an allometry for liana with DBH between 3 and 20 cm and then for lianas with DBH less than
3 cm, they used the allometry of early successional trees (di Porcia e Brugnera et al. 2019). Following a similar
approach and to make sure Palm has reasonable opportunity to compete with a reasonable diameter range, we assume
that the minimum height of Palm in the model is 4.8 m (corresponding to 10 cm DBH of Palm; other PFTs have a
minimum height of 1.5 m for recruitment), and when Palm grows to a height of 18 m (corresponding to 20 cm DBH)—
maximum height observed for the Palm in the forest (Figure 1)—they will allocate all the carbon to reproduction
instead of growth (relative allocation to reproduction is 1 for Palm, and 0.3 for other PFTs) (Table S1).
For other allometric relationships, such as leaf biomass-DBH, structural biomass-DBH, and crown area-DBH
relationships, we used the model default for Early, Mid, and Late PFTs, and assumed that Palm has the same
relationships as Early (Figure S1).

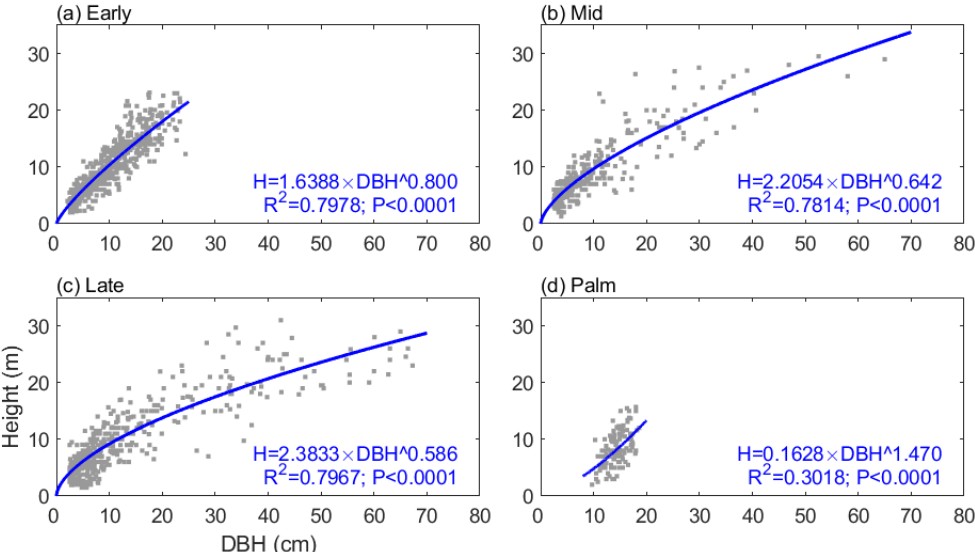

**Figure 1.** The height-diameter (DBH) relationship for the four PFTs: (a) Early, (b) Mid, (c) Late, and (d) Palm. The gray dots are
observations with outliers removed (Supplementary Information S1) and the blue lines are the estimated height-DBH relationship
based on these observations. The height-DBH model and the corresponding coefficient of determination ($R^2$) and p-value for each
PFT are given at the bottom of each panel.

### 2.2.3    Implementing Hurricane Disturbance

The ED2 model accounts for several types of disturbances, such as fires, land use, logging (Albani et al. 2006; Longo
et al. 2019a), but not hurricane disturbance. To account for hurricane impacts, we implement a hurricane-induced
wind mortality module and a seedling recovery module in the model. The wind mortality module consists of two
parts—the disturbance rate of the forest area ($\lambda_d$) and the survivorship of each cohort ($s_c$) in the disturbed areas. For
any patch with pre-disturbance area $A$, the area that is affected by disturbance ($A_d$) is proportional to $\lambda_d$, following
Moorcroft et al. (2001): $A_d = A\ [1 - \exp(-\lambda_d \Delta t)]$. The disturbed area ($A_d$) will be disturbed and become a new patch
(age 0), and the population within the new patch will be determined by the survivorship to disturbance. The remaining
area ($A - A_d$) will remain undisturbed, and the stem density will remain unchanged. The survivorship of each cohort
($s_c$) is the ratio of the cohort density that survived after the disturbance to the cohort density before the disturbance,
and it is cohort dependent. The cohorts that survived in disturbed areas will make up the new patch (age 0). In this
study, we assume that the forest is fully disturbed and $\lambda_d = 1$. The survivorship of each cohort $s_c$ is calculated as $s_c =$
$1 - \lambda_c$, where $\lambda_c$ is the mortality of each cohort. Based on previous analyses, $\lambda_c$ varies with hurricane strength, forest
structure, the PFT category and the DBH size of the cohort (Zhang et al. 2022b). First, we implement a binary model
for the mortality with respect to hurricane wind, where mortality occurs when hurricane wind exceeds a threshold and
no mortality otherwise. This binary model is built on the binary relationship between hurricane-induced forest damage
and hurricane wind speed from nine hurricane events at BEW between 1989 and 2017 (Supplementary Information
S2, S3, and S4). The wind speed threshold was set at 41 m s⁻¹ because the strongest hurricane wind that caused no
damage to the forest at BEW was 40 m s⁻¹ from hurricane Georges in 1998 and the lowest wind speed that caused
damage to the forest was 42 m s⁻¹ from hurricane Maria in 2017 (Supplementary Information S2, S3, and S4). If
mortality occurs (i.e., wind speed exceeds the threshold), the mortality rate of each cohort ($\lambda_c$) is a continuous function
of the size structure of the forest, represented by the proportion of large stems (DBH ≥ 10 cm) to the total recruited
stems (DBH ≥ 2.5 cm). Figure 2 shows the mortality of each PFT and DBH class during two hurricane events (Hugo
and Maria) based on census observations at BEW (see Section 2.1). We fit a logistic function to the mortality-structure
pair of each PFT and DBH class based on the observed pairs of mortality and structure from the two hurricane events.

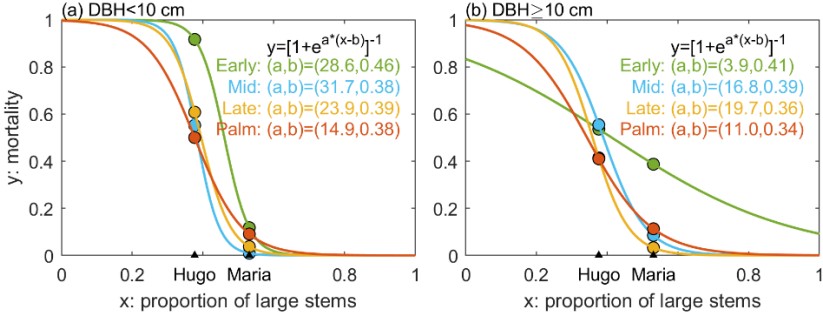


**Figure 2.** The mortality as a function of the size structure of the forest for each PFT and DBH class. The size structure is represented
as the proportion of large stems (DBH ≥ 10 cm) to the total number of stems in the forest (DBH ≥ 2.5 cm). The dots represent
observed mortality and proportion of large stems pairs from hurricane Hugo and hurricane Maria (Zhang et al. 2022b). Four colors
represent four PFTs. The solid lines represent the estimated mortality as a logistic function of the proportion of large stems. The
panel on the left is for small stems and that on the right is for large stems.

Hurricanes not only cause immediate stem mortality, but also affect the establishment of seedlings by opening
the canopy (Everham 1996; Brokaw 1998; Uriarte et al. 2009; Uriarte et al. 2012). Brokaw (1998) pointed out that
hurricanes promote germination and seedling establishment of the early successional species *C. schreberiana*, and
that the seedling establishment ends shortly after the disturbance as the canopy closes. The census data at BEW also
show abundant recruitments of the Early PFT in the first 20 years after hurricane Hugo and decreasing recruitment
with time (Zhang et al. 2022a). Therefore, based on the recruitment of Early PFT from the census data (Zhang et al.
2022a), we implement a recovery module where the seedling density from seed rain ($n_s$; individuals m$^{-2}$ yr$^{-1}$) decreases
with time since the last disturbance, and the reduction varies with PFT categories as:

$$n_s = n_0 \exp(-\alpha t),\tag{2}$$

where $n_s$ is the seedling density $t$ years after last hurricane disturbance, $n_0$ and $\alpha$ are PFT-dependent parameters.
Specifically, Mid, Late, and Palm PFTs maintain a low but constant seedling density ($n_0 = 0.05$ individuals m$^{-2}$ yr$^{-1}$
and $\alpha = 0$ yr$^{-1}$). The Early PFT has high seedling density ($n_0 = 0.2$ individuals m$^{-2}$ yr$^{-1}$) shortly after a hurricane
disturbance and the seedling rate decreases to the same value as other PFTs about 20 years after the disturbance ($\alpha =$
$0.06$ yr$^{-1}$), and it continues to decrease thereafter (Figure 3).

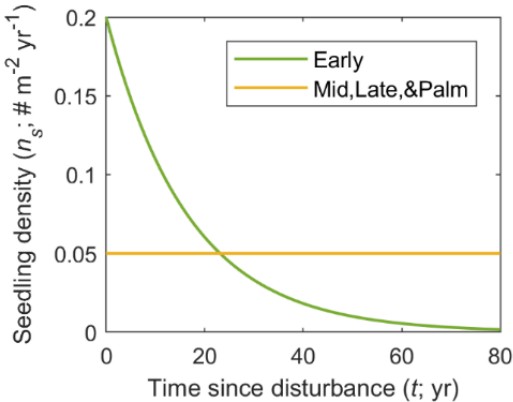


**Figure 3.** The seedling density from seed rain for each PFT as a function of time since disturbance.

### 2.3    Model Calibration and Validation

#### 2.3.1    The GLUE approach

The concept of Generalized Likelihood Uncertainty Estimates (GLUE) (Binley and Beven 1991; Beven and Binley
1992; Mirzaei et al. 2015) has been widely used to calibrate parameters in complex hydrological models. The steps of
GLUE include 1) generating a number of samples of the parameter set from a prior distribution of the parameters, 2)
running the simulation for each parameter set, 3) choosing a likelihood function (or weight function) to calculate the
weight of each simulation based on observations and the estimated outputs from the model simulation, and 4) selecting
the optimal parameter set and estimating the posterior distribution of the parameters and the posterior distribution of
the output variables. Here we use GLUE, for the first time, to calibrate the parameters in the ED2 model.

To obtain the prior distribution of parameters, we build on a previous parameter sensitivity analysis using the

ED2 model for a nearby forest in Puerto Rico by Feng et al. (2018). They demonstrated that model simulations are
sensitive to ten parameters, listed in Table 1, and provided the posterior mean and 95% confidence limits of the
parameters calibrated from plant traits observations using the Predictive Ecosystem Analyzer (PEcAn; LeBauer et al.
2013). We select the same parameters and use the posterior distribution of those parameters from Feng et al. (2018)
as the prior distribution for the GLUE in our study. We cannot just use their parameter distributions as final results
because our implementation has a site-specific set of allometric equations, explicitly represents palms as a separate
PFT and considers hurricane disturbances (Section 2.2). Feng et al. (2018) reported only the mean and the upper and
lower 95% confidence limits of the parameters (not the entire distribution), we assume that the parameters have
lognormal distributions. For the Palm PFT, we assume that it has the same distributions as Late, except that the woody
tissue density of Palm has the same distribution as that of Early.  From a different study system, Wang et al. (2013)
constrained the dark respiration factor from 0.01–0.03 to 0.01–0.016 by assimilating observations of model output
variables. Following Wang et al. (2013), we restrict the dark respiration factor to a smaller range with a uniform
distribution between 0.005 and 0.0175 for each PFT. Consistent with Meunier et al. (2022), we found that model
results are also sensitive to the parameter clumping factor (Figure S2). Therefore, we add the parameter of clumping
to the set being calibrated. Clumping factor is the ratio of effective LAI to the total LAI and affects the transmission
of radiation (Chen and Black 1992); it ranges from zero to one with zero representing leaves clumped in a single point
(0-area) and one representing leaves uniformly distributed in the unit area. Because of tree crowns, branches, and
subbranches, leaves of plant canopy are not uniformly distributed per unit area nor clumped at a single point. We
assume that the clumping factor is the same for all PFTs and the distribution of the clumping factor is uniform between
0.2 and 0.8.

We sample 10,000 realizations for the 41 parameters (10 parameters for each of the four PFTs and the one

clumping parameter for all PFTs) using the Latin Hypercube Sampling method embedded in MATLAB (Stein 1987).
We initialize the model with the pre-Hugo 1989 observations and run the model for 29 years, corresponding to 1989–
2018. The first 25 years (1989–2014) are used to calibrate the model with observations and the last four years (2015–
2018) for validation. We tested different calibration lengths (1989–1999, 1989–2004, and 1989-2009). 1989–2009
calibration period gives the same optimal simulation as 1989–2014 calibration period (Figure 4), but shorter
calibration lengths 1989–1999 (Figure S3) and 1989–2004 (Figure S4) throw away critical recovery information and
cannot give robust simulation in the validation period. We calculate the mean squared errors (*MSE*) of each realization
($j$, $j$=1, 2, …, 10,000) for the calibration period,

$$MSE_j = \frac{1}{nm}\sum_{t=1}^{m}\sum_{i=1}^{n}\left(\frac{X_{i,t,j} - Y_{i,t}}{\frac{1}{m}\sum_{t=1}^{m}Y_{i,t}}\right)^2 , \tag{3}$$

where $X_{i,t,j}$ represents the $j^{th}$ model simulations for variable $i$ at time $t$, and $Y_{i,t}$ represents observations for variable $i$ at
time $t$. The variables used to calculate *MSE* are stem density (individuals $m^{-2}$), average DBH growth rate (cm $(5yrs)^{-1}$),
and basal area (BA) ($cm^2\ m^{-2}$) for the four PFTs ($n$=12) (Figure 4). Times are the six census years ($m$=6) with
observations before hurricane Maria: post-Hugo 1989, 1994, 1999, 2004, 2009, 2014. Because BA is directly
calculated from the DBH of each cohort and weighted by the stem density of the cohort, the size structure (distribution
of stem DBHs) of the forest is implicitly represented with the variables overall stem density and total BA. Moreover,
the PFT composition is explicitly represented with the PFT-specific variables. Therefore, the *MSE* metric implicitly
measures the performance of a realization in describing the observed time series of the forest's size structure and PFT
composition.

We select the simulation with the smallest *MSE* as the optimal simulation and the corresponding parameter

set as the optimal parameter set. To obtain the posterior distribution of parameters, we first calculate the weight
(likelihood) of each realization following Binley and Beven (1991),

$$w_j = MSE_j^{-K} , \tag{4}$$

which is then rescaled to sum to one ($w_j/\sum_{j=1}^{N}w_j$), where $K$ is the parameter that controls the weight of each
realization. When $K = 0$, every simulation will have equal weights and when $K = \infty$, the single best simulation will
have a rescaled weight of 1 while all others being zero. We select $K$ such that the weighted standard deviations from
simulations are within and overlap as much as possible with the standard deviations of observations, indicating that
the parameters in those weighted simulations are reasonable given the uncertainty of the observations (Freer et al.
1996). The weighted standard deviation of variable $X$ is calculated as

$$\sigma_X = \sqrt{\sum_{j=1}^{N} w_j \left( X_j - m_X \right)^2} \,, \tag{5}$$

where $m_X = \sum_{j=1}^{N} w_j X_j$ is the weighted mean of the simulated variable. We find that $K=8$ has the best performance
on the posterior estimates of output variables stem density, aboveground biomass, basal area, proportion of each PFT,
and proportion of large stems (Figure 4, Figure S5, and Figure S6). Lastly, the posterior empirical cumulative
distribution function (CDF) of the parameters is obtained as

$$F(P \leq p) = \sum_{j:P_j \leq p} w_j \,. \tag{6}$$

The posterior empirical CDFs are then fit to lognormal distributions.

### 2.3.2    Non-Hurricane Mortality

The non-hurricane mortality of Palm is not well represented in the model (Figure S7), as initially calibrated. The
observed non-hurricane mortality is an overall mortality regardless of the cause of the death and is calculated from
non-hurricane censuses, whereas the non-hurricane mortality in model simulations includes aging mortality,
competition mortality, and disturbance mortality. We turned off all disturbances except for hurricane disturbance and
treefall disturbance. The disturbance mortality includes the background exogenous mortality and treefall disturbance
rate. Background mortality rate is 0.014 yr$^{-1}$ for small trees and zero for large stems because, following Moorcroft et
al. (2001), this mortality is accounted for in the treefall disturbance rate (i.e., the background mortality of large trees
is what causes the treefall disturbance). The treefall disturbance rate mortality is a combination of the area impacted
by treefall disturbance and the survivorship of this disturbance. By default, in ED2, it is assumed that the treefall
disturbance rate is 0.014 yr$^{-1}$, survivorship to treefall disturbance is zero for large trees and 10% for small trees, and
thus overall treefall mortality is 0.014 yr$^{-1}$ for large trees and 0.0126 yr$^{-1}$ for small trees. Competition mortality is
related to carbon starvation (i.e., negative net carbon accumulation) due to light and water limitation and varies with
cohorts. Aging mortality is the reciprocal of the longevity of the cohort without any biotic and abiotic influences, and
it is modeled as a constant for each PFT depending on the wood density of the PFT ($\rho_{PFT}$) relative to the wood density
of the Late PFT ($\rho_{Late}$): $0.15 \times (1 - \rho_{PFT}/\rho_{LATE})$ (Moorcroft et al. 2001). Since Palm has a much lower "wood" density
(0.31 g cm$^{-3}$; Swenson and Umana 2015) than the Late PFT (model default 0.9 g cm$^{-3}$), the aging mortality of Palm is
~0.1 yr$^{-1}$, or the longevity of palms would be equivalent to ~10 years. However, this is in contrast to the average age
of the palm species in the Luquillo Experimental Forest, which was found to be 61.1 years and the oldest palms were
more than 100 years old in 1982 (Lugo and Rivera Batlle 1987). This suggests that the aging mortality of Palm
calculated from its woody tissue density is a drastic overestimation. Therefore, we assume that the aging mortality of
Palm is independent of its woody tissue density and is 0 yr$^{-1}$, same as that of Late.

With a lower mortality (decreasing aging mortality from ~0.1 to 0), the density of Palm increases

continuously in the forest because of continuously recruiting seedlings, while the density of other PFTs and the AGB
of all PFTs are less affected (Figure S8). A previous study showed that hurricane disturbance can result in an increase
in seed production in the palm species (Gregory and Sabat 1996). Therefore, we calibrate the seedling recovery module
of Palm that we implemented in Section 2.2.3. Specifically, we test several recovery seedling densities (Eq. (2)) for
Palm, assuming that the seedling density of Palm is similar to that of Early—decreasing with time since disturbance—
but with different starting seedling level ($n_0$) and decaying factor ($\alpha$). We tested 36 combinations of $n_0$ varying from
0 to 0.05 individuals m$^{-2}$ yr$^{-1}$ with interval 0.01 individuals m$^{-2}$ yr$^{-1}$ and $\alpha$ varying from 0 to 0.05 yr$^{-1}$ with interval 0.01
yr$^{-1}$. We found that five of them lead to a smaller *MSE* (Eq. (3)) than the GLUE optimal simulation (0.1678, 0.1662,
0.1642, 0.1646, and 0.1691 for the five experiments and 0.1803 for the GLUE optimal), and the five combinations
have the same starting seedling density ($n_0$=0.02 individuals m$^{-2}$ yr$^{-1}$) but different values of the decaying factor
($\alpha$=0.01, 0.02, 0.03, 0.04, and 0.05 yr$^{-1}$, respectively) (Figure S9). To choose from the five decaying values, we
compared the recovery density schemes with the observed recruitment of Palms (stems entering the census with DBH
$\geq$ 2.5 cm and H $\geq$ 1.5 m each year). As we do not have seedlings but only recruited stems in our census data, we
assumed that seedling density has the same response (varying with time since disturbance) as recruitment, but not
necessarily the same magnitude (density) as recruitment. Based on the census data, there were 37, 64, 50, 34, and 32
palms recruited in the 85 plots (78.5 m$^2$ each plot) in 1994, 1999, 2004, 2009, and 2014 censuses, respectively, which
corresponds to 0.0011, 0.0019, 0.0015, 0.0010, and 0.0010 individuals m$^{-2}$ yr$^{-1}$ after 5, 10, 15, 20, and 25 years of the
Hugo disturbance. In other words, the recruitment decreases to half of the starting level in 20–25 years, or a decaying
factor $\alpha$≈0.03 yr$^{-1}$. We assume that the seedling density has the same decaying rate as the recruitment density and thus
we select the seedling density scheme $n_0$=0.02 individuals m$^{-2}$ yr$^{-1}$ and $\alpha$=0.03 yr$^{-1}$ as the seedling recovery scheme
for Palm.

After changing the aging mortality of Palm to zero and the seedling density to a lower and slowly decreasing

value, we did not repeat the GLUE. This is because Palm has constrained DBH size (between 10 and 25 cm) and
decreasing the aging mortality increases its density while decreasing seedling reproduction decreases its density,
which maintains the overall density of Palm, without affecting other variables of Palm nor variables of other PFTs
(Figure S9). Therefore, we use the parameter set found from the GLUE (Table 1) but with 0-aging mortality and a
lower seedling density recovery ($n_0$=0.02 individuals m$^{-2}$ yr$^{-1}$ and $\alpha$=0.03 yr$^{-1}$) for simulations in the following studies.
**2.4    Parameter Sensitivity Analyses and Variance Decomposition**
Using a similar approach to PEcAn (LeBauer et al. 2013), we analyze the sensitivity of model simulations to the
parameters and the contribution of the parameters to the variances. Specifically, we set up nine experiments for each
of the 41 parameters, corresponding to the nine quantiles (10$^{th}$, 20$^{th}$, …, 90$^{th}$) of the posterior distribution of each
parameter, while all other parameters remain constant at their optimal. For the total 369 sensitivity experiments, we
initialize the model with the pre-Hugo observation and run each experiment for 25 years (1989–2014).

To study the stability of the optimal parameter set, we calculate the *MSE* of each experiment and compare it

with the *MSE* of the optimal. To quantitatively study the sensitivity of output variables to the parameters, we calculate
the standardized cubic regression coefficient ($\beta$),

$$\beta = \frac{\partial \tilde{x}(p_o)}{\partial p_o} \Big/ \frac{x_o}{p_o}, \tag{7}$$

where $p$ and $x$ are a specific parameter and the corresponding output variable. $\tilde{x}$ is the cubic regression function of $x$
on $p$: $\tilde{x} = ap^3 + bp^2 + cp + d$, estimated from the pairs of parameter $p$ and variable $x$ along the nine quantiles of the
posterior distribution of parameter $p$. $\frac{\partial \tilde{x}(p_o)}{\partial p_o}$ is the partial derivative of $\tilde{x}$ on $p$ at $p_o$, where $p_o$ and $x_o$ are the optimal
value of the parameter and the corresponding output variable. Only when the $R^2$ metric of the regression function is
significant at 99% confidence level via student-$t$ test is $\beta$ calculated. We calculate $\beta$ for 20 variables [stem density,
BA, AGB, and leaf area index (LAI) of each PFT and of all PFTs] and for the 41 parameters. The $\beta$ for the variables
at the first and the 25$^{th}$ simulation years are selected to represent the short-term and long-term response of modeled
variables to the parameters, respectively.
To quantitatively study the uncertainty of the simulated variables (stem density, AGB, BA, LAI, etc.) from
the uncertainties of the parameters, we calculate the coefficient of variation ($\theta$) for each variable resulting from
experiments with different parameters:

$$\theta = \frac{\sigma}{\mu}, \tag{8}$$

where $\sigma$ and $\mu$ are the standard deviation and the mean value of the variable from the nine experiments of the parameter.
To study the contribution of each parameter to the uncertainties of the simulated variables, we calculate the total
variance from all the sensitivity experiments ($Var_T$) and the variance from experiments of each parameter ($Var_p$), and
decompose the total variance as follows,

$$Var_T = \sum_{p=1}^{Np} Var_p + \omega, \tag{9}$$

where $Var_p$ is the variance of model outputs from experiments with different values of parameter $p$, and $Np$ is the total
number of parameters ($Np$=41), $\omega$ represents the variance from the interaction among parameters.
**2.5   Experiments with Different Initial Conditions**
To study the impact of the initial condition of the forest on the recovery, we set up two experiments with different
initial forest states (pre-Hugo state and pre-Maria state) with a hurricane disturbance in the first simulation year
(experiment IhugoH1 and experiment ImariaH1, hereafter), and one control experiment with pre-Hugo state and no
hurricane disturbance in all simulation years (experiment IhugoHn, hereafter). The three experiments run for 112
simulation years (corresponding to years 1989–2100). The meteorological drivers between 1989 and 2017 are
observations from meteorological towers at BEW, and the meteorological drivers between 2018 to 2100 are randomly
sampled from the observations between 1989 and 2017. Hurricane disturbance is turned off in all simulation years for
experiment IhugoHn and in all but the first simulation year for experiments IhugoH1 and ImariaH1. Thus, experiment
IhugoHn represents the succession of the forest without hurricane disturbances for more than a century. Experiments
IhugoH1 and ImariaH1 represent the recovery of the forest from a hurricane disturbance given different initial
conditions of the forest.
**3   Results**
**3.1   Model Assessment**
**3.1.1   Optimal Simulation and Optimal Parameter Set**
Figure 4 shows the optimal model simulation along with census observations for years 1989–2018. The simulated
stem density of Early increased from 0.0027 individuals m$^{-2}$ in 1990 to 0.0324 in 1994 (1100% increase) and to 0.0748
in 1999 (131% increase) and decreased steadily thereafter, consistent with observations (0.0030 individuals m$^{-2}$ in
post-Hugo 1989, 1673% increase in 1994 and 84% increase in 1999). The simulated stem density of Mid is overall
underestimated by 47% compared to the mean from the 85 plots of observations but is within one standard deviation
of the observations. The simulated stem density of Late and Palm are also within one standard deviation of the
observations although the mode predictions suggest 25% underestimation and 38% overestimation, respectively. The
optimal simulation overestimates the growth rate of the Early PFT by 133% for years between 2000 and 2014, but it
generally captures the decrease of growth rate with time since the hurricane disturbance for all PFTs. Furthermore,
the optimal simulation agrees well with the observations for the overall stem density (-21% relative bias), basal area
(-12% relative bias), and aboveground biomass (-15% relative bias), and captures well the PFT composition (+1%, -
8%, -2%, and +9% differences in the percentages of Early, Mid, Late, and Palm PFTs, respectively) and size structure
(+0.8% differences in the percentage of large stems) (Figure 5).

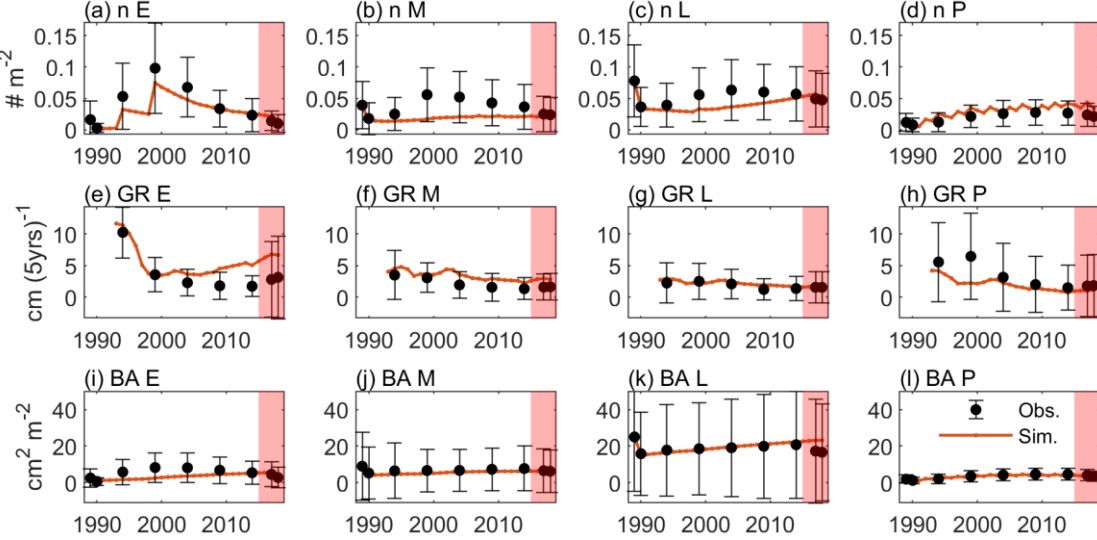


**Figure 4.** Time series of variables from observation (dots and error bars) and the optimal simulation (red lines). (a)-(d) stem density
of all trees (n; DBH ≥ 2.5 cm) (individuals m$^{-2}$) for Early, Mid, Late, and Palm PFTs, respectively. (e)-(h) diameter growth rate
(GR; cm (5yrs)$^{-1}$) for the four PFTs; (i-l) basal area (BA; cm$^2$ m$^{-2}$) for the four PFTs. The dots and the error bars represent the
means and the one standard deviations from the means across the 85 plots. Period between 1989–2014 is for model calibration and
period between 2015–2018 is for model validation (shaded).

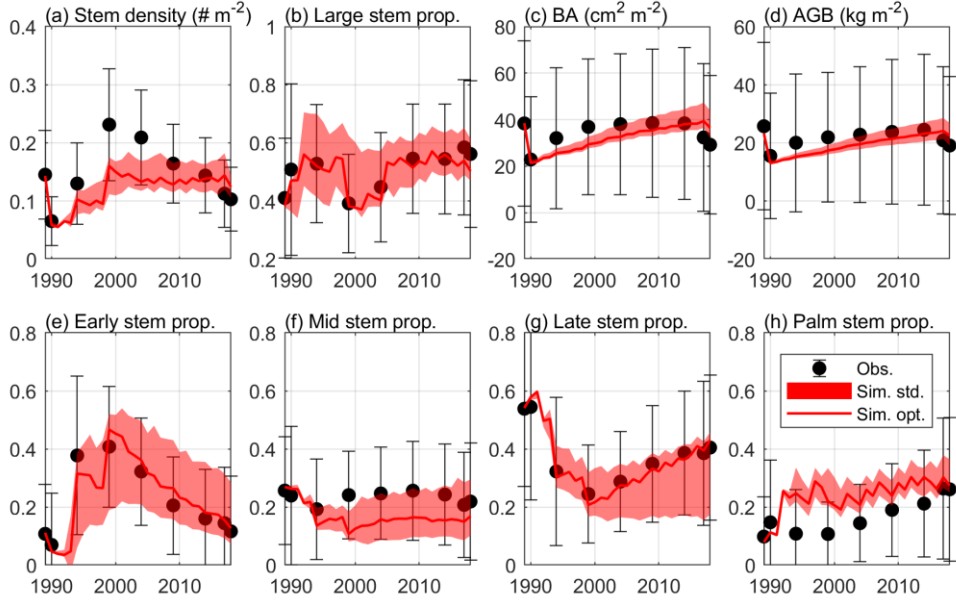


**Figure 5.** The standard deviation of the estimated variables with *K*=8 in equation (4), along with the optimal simulation and
observation. The figure shows (a) stem density of all stems with DBH ≥ 2.5 cm (individuals m$^{-2}$), (b) stem density proportion of
large stems with DBH ≥ 10 cm, (c) basal area (BA; cm$^2$ m$^{-2}$), (d) aboveground biomass (AGB; kgC m$^{-2}$), and stem density
proportion of (e) Early, (f) Mid, (g) Late, and (h) Palm PFTs.

In the verification period between 2015–2018, the simulated overall stem density, basal area, and
aboveground biomass have a relative bias of +24%, +23%, and +17%, respectively, compared to the mean of the
observations. The simulated percentages of the four PFTs have a difference of +3%, -7%, -4%, and 8%, respectively;
and the simulated large stem percentage has a difference of +0.3% compared to the mean of the observations. Overall,
the simulated variables between 2015–2018 are within the standard deviations of the observations (Figure 4 and Figure
5), suggesting that the parameters found using the data between 1989–2014 are valid for the 2015–2018.

Table 1 shows the optimal set of the parameter values. The clumping factor (0.34) is lower than that from
other studies in different locations (~0.7; He et al. 2012). Other parameters are reasonable and are consistent with
reported values. For example, the leaf turnover rate of Late (0.16 yr$^{-1}$) is consistent with a previous study (~0.1; Gill
and Jackson 2000). The leaf turnover rate of Palm (0.42 yr$^{-1}$) is consistent with previous observations of 0.36 yr$^{-1}$ at
BEW (Lugo et al. 1998). The woody tissue density of Palm (0.24 g cm$^{-3}$) is consistent with previous observations of
0.31 g cm$^{-3}$ (Swenson and Umana 2015).

**Table 1.** The optimal parameter set obtained from the GLUE method.

| Parameter Name | Units | Early | Mid | Late | Palm |
|---|---|---|---|---|---|
| clumping factor (Clf) | proportion | 0.34 | | | |
| fine root allocation (FRA) | ratio | 0.64 | 1.2 | 0.95 | 1.85 |
| leaf turnover rate (LTR) | yr$^{-1}$ | 1 | 0.83 | 0.16 | 0.42 |
| leaf width (LWd) | m | 0.1 | 0.07 | 0.16 | 0.13 |
| quantum efficiency (Qef) | mol$_{CO2}$ mol$^{-1}_{photon}$ | 0.055 | 0.069 | 0.038 | 0.05 |

| | | | | | |
|---|---|---|---|---|---|
| dark respiration rate (Rdf) | proportion | 0.0071 | 0.0144 | 0.0143 | 0.0088 |
| growth respiration rate (Rgf) | ratio | 0.44 | 0.595 | 0.421 | 0.401 |
| specific leaf area (SLA) | $m^2 kg^{-1}$ | 23.26 | 22.28 | 13.19 | 14.15 |
| stomatal slope (SSp) | ratio | 6.17 | 8.02 | 5.35 | 5.07 |
| carboxylation rate (Vm0) | $\mu mol_{CO_2}\ m^{-2} s^{-1}$ | 23.32 | 21.73 | 9.29 | 12.24 |
| wood density (WDe) | $10^3\ kg m^{-3}$ | 0.32 | 0.6 | 0.77 | 0.24 |


### 3.1.2    Posterior Distribution of Parameters

Figure 6 shows the posterior and prior probability distribution functions (PDFs) of the parameters. The most significant
differences between the posterior and the prior distributions are for the parameters of clumping factor (Clf) and dark
respiration rate (Rdf). The posterior PDFs of some parameters (i.e., carboxylation rate, specific leaf area, leaf width,
stomatal slope, and wood density), which are well constrained by observational trait data (Feng et al. 2018), do not
change much from the priors (the maximum difference between the prior and posterior CDFs is generally less than
0.1). The posterior PDFs of other parameters (e.g., leaf turnover rate, quantum efficiency, and fine root allocation),
especially for the Early and Mid PFTs, with few observational trait data (Feng et al. 2018), changed greatly from the
prior distributions (the maximum difference between the distributions is around 0.3).

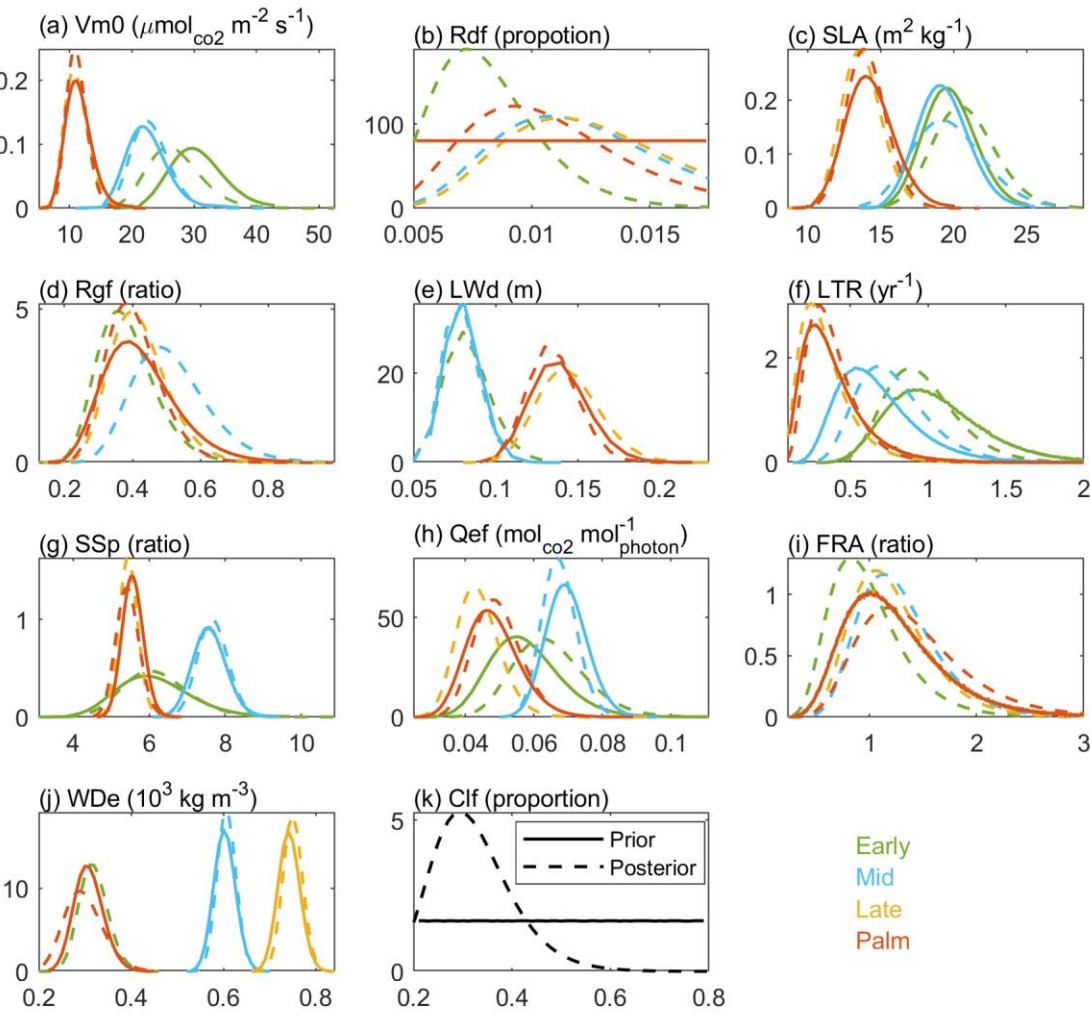

**Figure 6.** The prior (solid line) and posterior (dashed line) probability density functions for the four PFTs (colors) of the 11 parameters. The first ten parameters are PFT-dependent, and the last one leaf clumping factor (Clf) is PFT-independent. Palm has the same prior distribution as Late for all parameters except that the wood density (WDe) of Palm has the same prior distribution as that of Early. The long name of each parameter is shown in Table 1.

### 3.1.3 Parameter Sensitivity and Uncertainty

Among the 369 sensitivity experiments with different parameter values, 57 of them have slightly smaller *MSE*s than the optimal, but the simulated variables (stem density, AGB, PFT composition, and size structure) from those experiments are very close to those from the optimal (Figure S10), indicating that the optimal simulation we found from GLUE is stable given the uncertainties of the parameters.

In terms of the sensitivity of simulated variables on the parameters, the magnitude of standardized cubic regression coefficients ($\beta$) is generally low (~0.2) in the first simulation year (Figure 7 a), indicating that the parameters do not have a strong effect on the variables. LAI is the most sensitive variable in the short term, and it is sensitive to both the specific leaf area (SLA) of its own PFT and the clumping factor (Clf). Furthermore, each PFT is mainly

sensitive to the parameters of its own PFT, and vice versa (Figure 7 a). After 25 years of simulation, the sensitivity of
the variables on the parameters becomes more complex (Figure 7 b). First, the magnitude of $\beta$ increases significantly,
indicating that the parameters show stronger impacts on the variables in the long term. Second, the variables are
sensitive to different parameters in the short term and in the long term. For example, SLA and clumping factor are the
most important parameters to LAI in the first simulation year, but not after 25 years of simulation. Instead, quantum
efficiency (Qef) and dark respiration (Rdf) are the most important parameters to LAI after 25 years of simulation.
Third, besides the sensitivity of variables to the parameters of their own PFT, variables of a specific PFT also show
sensitivity to the parameters of other PFTs. For example, the variables of Early and Mid PFTs are not only sensitive
to Early and Mid PFTs parameters, but also sensitive to Late PFT parameters. Specifically, the quantum efficiency,
wood density, and specific leaf area have significant positive effects on the variables of its own PFT, but significant
negative effects on other PFTs. The Palm PFT is sensitive to its own parameters, but also to the specific leaf area of
the Early PFT (Figure 7 b).

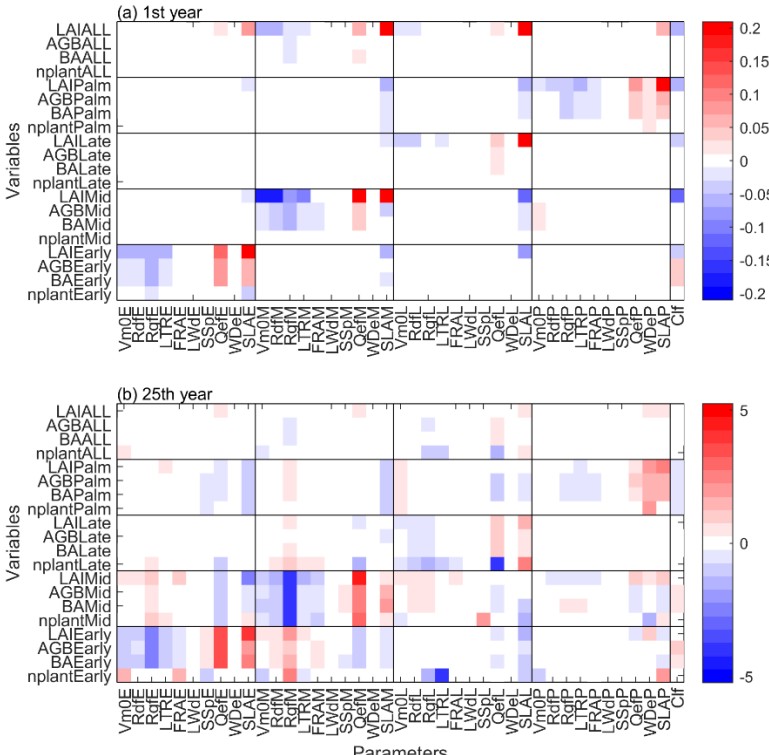


**Figure 7.** The standardized cubic regression coefficient ($\beta$) of variables at (a) first and (b) 25th year of the simulations regarding
the parameters. The variables include stem density (nplant), basal area (BA), aboveground biomass (AGB), and leaf area index
(LAI) for each PFT. The parameters include 10 PFT-dependent parameters and one PFT-independent parameter listed in Table 1.

The stem density has a larger variation than LAI, BA and AGB after 25 years of simulation (Figure 8). Given

that large stems contribute more to LAI, BA, and AGB, larger variation of stem density than LAI, BA, and AGB
indicates that small stems are more variable than large stems. The variation of those variables also varies with PFTs.
For the stem density, Late PFT has the largest variation, followed by Early, then Mid, and Palm has the smallest
variation, indicating that stem density of small Late is the most sensitive to the uncertainty of the parameters. For BA,
AGB, and LAI, Early and Mid PFTs show the highest variability, followed by the Palm PFT, and the Late PFT has
the lowest variation, indicating that large stems of Early and Mid PFTs are more sensitive to the uncertainty of the
parameters than large stems of Late and Palm PFTs.

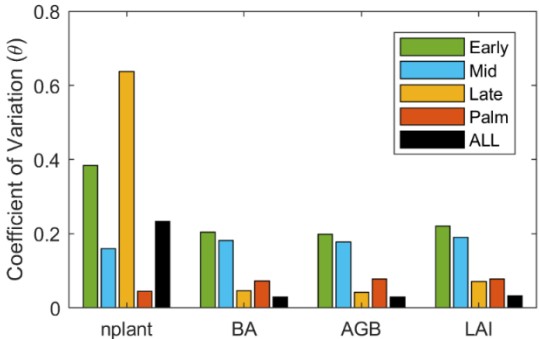


**Figure 8.** The coefficient of variation ($\theta$) for the variables of each PFT at the 25th simulation year.

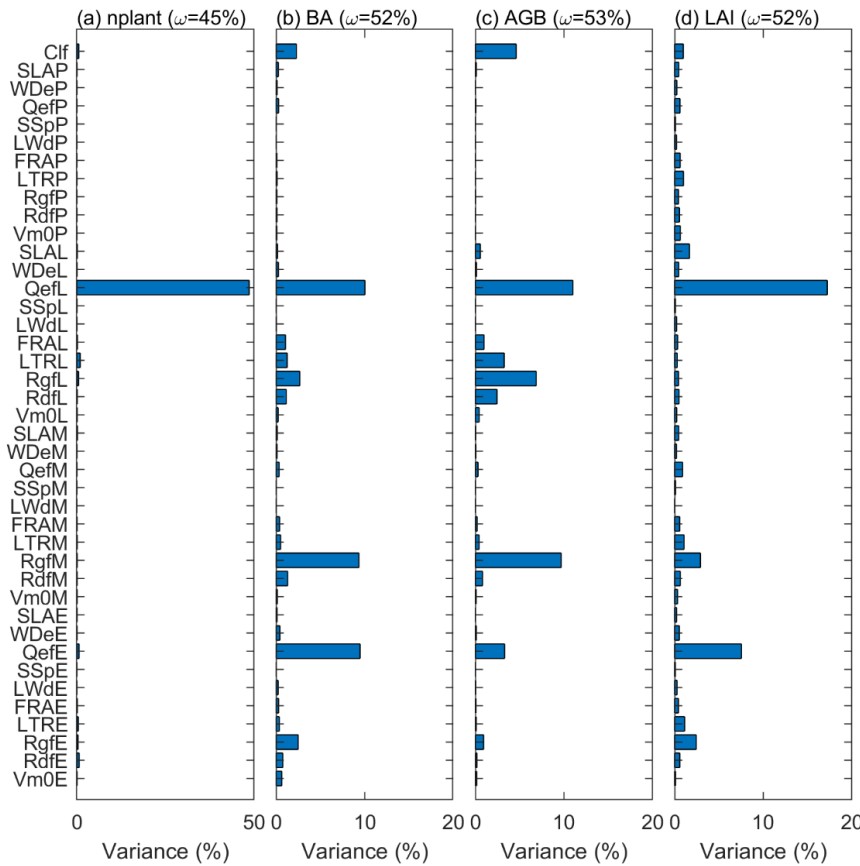


**Figure 9.** The variance explained by each parameter for variables (a) stem density, (b) basal area, (c) aboveground biomass, and
(d) leaf area index. The variance explained by the interaction among parameters are given in the parenthesis.

The variance decomposition analyses reveal that 50% of the uncertainty of the stem density comes from the

quantum efficiency of Late (QefL) (Figure 9). However, QefL explains less than 10% of the uncertainty in BA, AGB,
and LAI, indicating that QefL has significant effects on the density of small stems, but less effects on the density of
large stems. In other words, QefL impacts the recruitment and establishment of stems more than the growth of stems.
The uncertainty of the growth of stems comes from the growth respiration factor (Rgf), which explains about 10% of
the uncertainty. The interaction among parameters accounts for 21% of the uncertainty of the stem density, and more
than 50% of the uncertainty of the BA, AGB, and LAI.
**3.2    Impact of Initial Condition on Forest Recovery**
Figure 10 shows the 112-year simulations of the forest initialized with different forest states (pre-Maria state and pre-
Hugo state) with or without hurricane disturbance at the first simulation year. Without hurricane disturbance
(IhugoHn), the forest experiences a decrease (-17%) in stem density in the first 10 years due to the self-thinning
process of the forest (Figure 10 a). The decrease is mainly attributed to mortality of small stems of Mid and Late PFTs
(Figure S11 b and c), which leads to an increase (5%) in the proportion of large stems (DBH $\geq$ 10 cm) (Figure 10 b)
but BA and AGB remain steady (Figure 10 c and d). After 10 years, a large number of Early PFT stems recruit with
DBH less than 10 cm (Figure S11 a), decreasing the overall large stem proportion. After 30 years, Mid trees recruit
and grow (Figure S11 b and Figure S12 b), increasing the total BA and AGB (Figure 10 c and d). As small Late trees
recruit frequently after 20 years (Figure S11 c), the stem density increases steadily, and the proportion of large stems
decreases steadily. Because small stems contribute little to BA and AGB, BA and AGB have a slower increase with
time (Figure 10 c and d) than stem density (Figure 10 a).

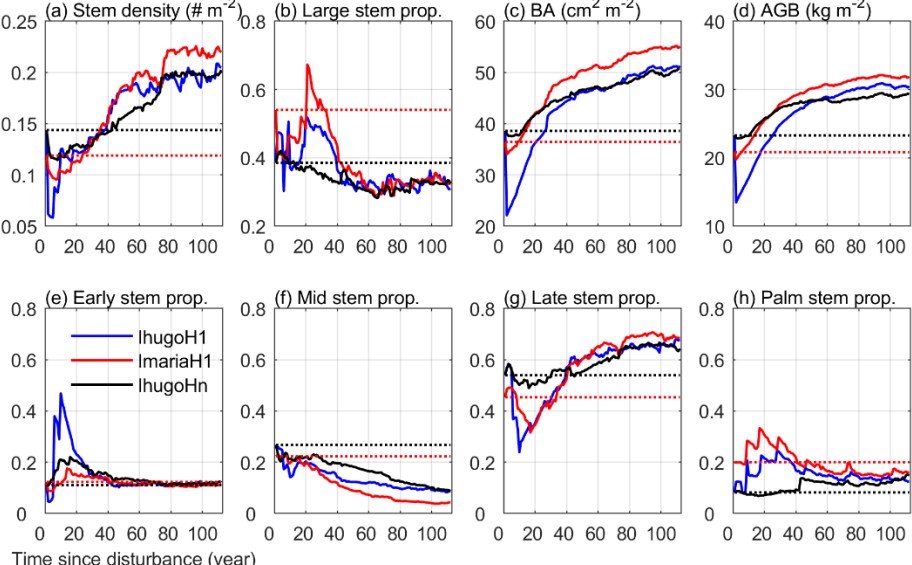


**Figure 10.** Time series of eight variables from the simulation of the three experiments: IhugoHn, IhugoH1, ImariaH1. The dotted
lines are the initial state of the variables for each experiment (IhugoHn and IhugoH1 have the same initial state). The variables in
(a) stem density, (c) basal area, and (d) aboveground biomass are for stems with DBH $\geq$ 2.5 cm. The stem proportion in (b) is the
proportion of the stem density with DBH $\geq$ 10 cm to the stem density with DBH $\geq$ 2.5 cm. The variables in (e)-(h) are the proportion
of the stem density of each PFT with DBH $\geq$ 2.5 cm to the total stem density of all PFTs with DBH $\geq$ 2.5 cm.

After 80 years, the PFT composition reaches a steady state (the change of 30-year moving average is less
than 1% compared to the previous year; Figure S13), where the Early, Mid, Late, and Palm PFTs account for 11.8%,
10.6%, 65.3%, and 12.3% of the total stem density, respectively (Figure 10 e, f, g, h). This state is significantly
different from the initial state and exhibits a 16% reduction on the proportion of the Mid PFT. It exhibits increases on
all other PFTs proportions (+0.7%, +11.4%, and +4.1% for Early, Late, and Palm, respectively). The Early PFT has
stems of all DBH classes (Figure S11 a); while Mid PFT has mostly small stems with DBH less than 5 cm and a small
cohort (2 individuals ha$^{-1}$) of large stems with DBH around 200 cm (Figure S14 b and f), which contributes a
significant portion to the total AGB (Figure S12 b). The Late PFT is the most abundant PFT (Figure S11 c) and
contributes the most to the total AGB in the forest (Figure S12 c). The stem density of Late decreases with DBH
(Figure S11 c), and the largest-DBH cohort reaches 180 cm (Figure S14 c), which is smaller than that of Mid but has
a higher density (7 individuals ha$^{-1}$) (Figure S14 g). The maximum DBH is far larger than that we observed (89 cm in
2017), which could be an overestimation due to no nutrient limitation. Palm recruits with DBH between 10 and 15
cm, the DBH grows slowly after recruitment, and DBH growth stops after they reach the reproduction height (18 m,
and 25 cm in DBH correspondingly) and allocate all carbon to reproduction (Section 2.2.2), hence palms do not exceed
25 cm DBH (Figure S14 d) and most of them are between 10 and 20 cm (Figure S11 d and Figure S12 d). This is in
agreement with the maximum reported values of DBH (Lugo and Rivera Batlle 1987).
Compared with the experiment without hurricane disturbance in the first simulation year (IhugoHn), the
experiments with hurricane disturbance in the first simulation year (IhugoH1 and ImariaH1) reach higher BA and
AGB levels after 60 years of succession from the hurricane disturbance (Figure 10 c and d). This is due to the carbon
accumulation of large Late PFT in disturbed forests (Figure S12 g and k). Large Late trees in disturbed forest (IhugoH1
and ImariaH1) have higher growth rate and lower background mortality rate compared to those in the undisturbed
forest (IhugoHn) (Figure 11) because of the decreased competition to reach the open canopy. As the disturbed forest
recovers, the BA and AGB increase to the level of the undisturbed forest (Figure 10 c and d), the growth rate decreases
(Figure 11 a) and the mortality rate increases to the levels of those in the undisturbed forest, especially for severely
disturbed forest (IhugoH1) (Figure 11). With lower mortality and higher growth rate in the first 60 years, there will
be more large Late trees in the canopy at the end of the simulation (12 individuals ha$^{-1}$ vs 8 individuals ha$^{-1}$) (Figure
S14 g) even though the maximum DBH will be smaller (Figure S14 c).

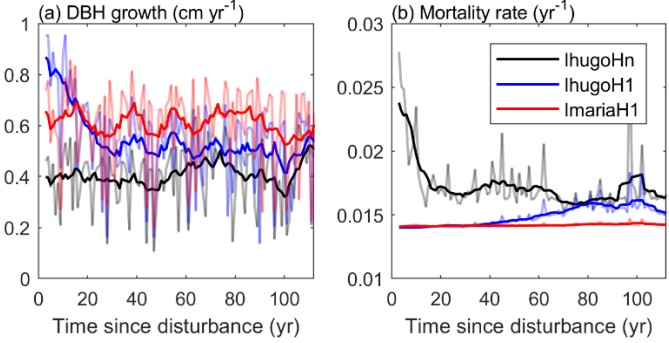

**Figure 11.** Times series of (a) average growth rate and (b) mortality rate of Late trees with DBH ≥ 20 cm. The light-colored lines
represent the yearly values, and the solid lines are ten-year moving averages.

The recovery is different with different initial states. With pre-Hugo state (IhugoH1), the forest takes 25 years
to recover to the pre-disturbance BA and AGB levels (Figure 10 c and d), but with pre-Maria state (ImariaH1), it takes
only 10 years to recover to the pre-disturbance BA level (Figure 10 c) and 5 years to the pre-disturbance AGB level
(Figure 10 d). The succession dynamics are different, too. With pre-Hugo state, the hurricane-induced mortality is
very high, and thus the canopy opens, and Early and Palm PFTs recruit greatly in the first 20 years (Figure S11 e and
h), and then it is taken over by the Late PFT (Figure S11 g). With pre-Maria initial state, the hurricane-induced
mortality is low, and the canopy is not significantly changed after the hurricane, and Early PFT does not recruit as
much as it does in the pre-Hugo state initialized simulation (Figure S11 i and e). The PFT composition after 100 years
is similar for the two simulations, but the BA and AGB are not (Figure 10). The BA and AGB with the pre-Maria
initialization are higher than those with the pre-Hugo initialization throughout the 110 years of simulations, even
though the initial BA and AGB levels in the pre-Maria state are lower than those in the pre-Hugo state (Figure 10 c
and d). This is because of the higher mortality at the first year with pre-Hugo state, leading to a larger reduction in the
density of large stems. With the succession following the disturbance, there are more large stems, especially Late and
Palm, in the pre-Maria simulation than in the pre-Hugo simulation (Figure S14), contributing to the higher AGB and
BA in the pre-Maria simulation (Figure S12 g, h, k, and l).
**4    Discussion**
We developed a hurricane module (including a mortality module and a recovery module) for the ED2-HuDi model,
based on census observations. We then applied a parameter estimation algorithm, GLUE, to calibrate important
parameters in the model and selected the optimal parameter set for the final model simulation. However, because the
observations are limited to only two hurricane events, the hurricane module may be biased toward the two
observations. The simulation results show some discrepancies with observations, and these discrepancies could be in
part due to the GLUE approach and parameter uncertainties. Here we discuss the uncertainty associated with the
developed hurricane module, the limitations and advantages of the GLUE framework, and the uncertainties of model
outputs.
**4.1    Uncertainty of the hurricane module**
We included a hurricane mortality module and a hurricane recovery module for hurricane disturbance. Crown damage
is also an important part of hurricane disturbance and could have important impact on forest structure and carbon
accumulation (Leitold et al. 2021), but we did not include crown damage in the hurricane disturbance module because
the census data used to develop and calibrate the module do not include crown damage information. The hurricane
mortality module was developed based on observations from two hurricane events at the study site. The relationship
between mortality and forest size structure (proportion of large stems) was fitted to a logistic function (Figure 2) for
each PFT and DBH class. Generally, Palm PFT has a lower mortality than other PFTs, but Palm mortality was higher
(11% for Palm, 9% for Mid, and 3% for Late) when the forest was dominated by large stems (e.g., large stem
proportion is 0.6, except for the high mortality of 39% for Early (Figure 2b). This was due to the high mortality of
Palm during Maria, which was a result of plant pathogens (Zhang et al. 2022b; Heartsill Scalley 2017). The mortality
of large-stem Early PFT is significantly different from other PFTs, and this difference was due to the significantly
higher mortality of large-stem Early during hurricane Maria compared to other PFTs. Such high mortality of large-
stem Early may be a result of other factors besides hurricane disturbance, and it could be further studied if there were
more observations. Future work could include observations from other study sites to improve the hurricane disturbance
module.

There are four critical parameters associated with the hurricane disturbance module, including disturbance

rate of forest area ($\lambda_d$) and survivorship of each cohort ($s_c$) from the mortality module, initial seedling density ($n_s$) and
decay factor of seedling density with time since disturbance ($\alpha$) from the recovery module. We tested the sensitivity
of the parameters of the recovery module but did not test the uncertainty of the parameters of the mortality module
because the values are from observations at the study site. For future studies using this module, either testing the
uncertainty of the parameters or using site specific values are encouraged.
**4.2**     **Limitations and Advantages of GLUE**
GLUE samples from continuous distributions, but the sampled parameter sets are in a discrete space, therefore, the
GLUE approach may not lead to the true optimum due to the finite number of samples. To justify the sample size of
10,000 for 41 parameters in this study, we repeated GLUE for a larger sample size (20,000). The optimal simulation
from 20,000-sample GLUE (Figure S15) is very similar to that from the 10,000-sample GLUE (Figure 4) and the
optimal parameter sets from the two GLUEs are similar, suggesting that the two GLUEs found an optimum around
the same local optimum and 10,000 samples are sufficient for the 41 parameters. However, given the nature of
equifinality, there may be multiple parameter sets that can lead to the same observed state (Beven and Freer 2001),
and thus the optimal parameter set we found from GLUE may be one of many possible solutions.

Although GLUE may not guarantee the global optimum, it implicitly handles any effects of model

nonlinearity, model structure errors, input data errors, and parameters covariation (Beven and Freer 2001). Moreover,
GLUE allows us to optimize parameters using any variables of interests in the cost function. For example, in our study,
we want to make sure the model captures the size structure and PFT composition of the forest community, and thus
we utilized forest stand variables including stem density, growth rate, and BA of each PFT in the cost function.
Compared to other optimizers (such as PEcAn) that calibrates parameters using plant traits observations (e.g., wood
density, leaf turnover rate) before running model simulations, GLUE's ability of constraining parameters from model
output variables utilizing observations of forest stand variables (BA, AGB, etc.) could further reduce the uncertainty
of parameters (Wang et al. 2013). Note that we did not calibrate the parameters using plant traits observations in this
study, because the parameters we use are already calibrated with plant traits observations in Feng et al. (2018) and we
adopted their calibrated parameters in our study (see Section 2.3.1).
**4.3**     **Uncertainty of Model Outputs from Parameters**

To be consistent with census observations, we included stems with DBH $\geq$ 2.5 cm in the analyses. The large

variation of simulated stem density (Figure 8) could be due to the timing of cohorts exceeding the 2.5 cm threshold,
and thus can be minimized by averaging stem density over several years (Massoud et al. 2019). The optimization is
sensitive to light-related parameters, such as clumping factor, quantum efficiency, and dark respiration (Figure 9).
This is possibly because light limitation is the most important limitation in the model, as water is not limited in this
tropical site, and we turned off nutrient limitation. This is consistent with Meunier et al. (2021) who found that light
limitation contributes partly to model uncertainties. The clumping factor we calibrated for our study site is lower than
that from other locations (He et al. 2012), which could be due to uncertainties of the allometries and estimates on the
Leaf Area Index (LAI). LAI is generally underestimated in the vegetation dynamics models (e.g., Xu et al. 2016). As
discussed in Shiklomanov et al. (2021), the ED2 model has a less robust estimation on LAI because of structural errors
in representing direct radiation backscatter. Both LAI and the clumping factor are rarely measured, and LAI estimated
from satellite remote sensing data often have variable quality, especially in tropical forests (Xiao et al. 2016, 2017).
Future census practices should include LAI and the clumping factor. Even though the LAI measured from the ground
may be different from the LAI measured from above the canopy (with airborne lidar or satellites), ground
measurements could provide useful information for both the vertical structure of the forest and the quality of satellite
remote sensing and airborne lidar data. Furthermore, acclimation to understory light is not considered in this model,
however, traits respond strongly to light environments (Lloyd et al. 2010; Keenan and Niinemets 2016), therefore it
needs to be considered in future developments (Xu and Trugman 2021).
Our results that modeled variables have different responses to parameters in the short term (e.g., first
simulation year) and in the long term (e.g., $25^{th}$ simulation year) agree with a previous study (Massoud et al. 2019).
Furthermore, we showed that variables of a specific PFT are most sensitive to the parameters of the same PFT, but
also sensitive to parameters of other PFTs. Those interactions between variables and parameters indicate the
competition among PFTs. For example, Palm is sensitive to its own parameters, but also to Early SLA. This can be
explained by the competition for light between Early and Palm, where a higher SLA of Early PFT leads to a higher
LAI of Early allowing Early to photosynthesize more efficiently and thus be more competitive in the community.
Those competitions are important for the co-existence of PFTs in model simulations and critical to the PFT
composition and succession.
**5    Conclusion**
Hurricanes are a major disturbance to tropical forests, but hurricane disturbance had not been implemented in any
model of vegetation dynamics. In this study, we implemented hurricane disturbance in the Ecosystem Demography
model (ED2) and calibrated the model with forest stand observations of a tropical forest in Puerto Rico. The calibrated
model has good representation on the recovery trajectory of PFT composition, size structure, stem density, basal area,
and aboveground biomass of the forest. We used the calibrated model to study the recovery of the forest from a
hurricane disturbance with different initial forest states and found that a single hurricane disturbance changes forest
structure and composition in the short term and enhances AGB and BA in the long term compared with a no-hurricane
situation. Forests with wind-resistant initial state will have lower mortality, recover faster, and reach a higher BA and
AGB level than forests with a less wind-resistant initial state.
The model developed and results presented in this study can be utilized to understand the fate of tropical
forests under a changing climate. Hurricanes are likely to become more frequent and severe in the future with global
warming (IPCC 2021). With frequent hurricane disturbances in the future, forests will not have enough time to reach
a steady state, and the structure and composition will be constantly changing, which provides different initial states
for future hurricane disturbances and thus different recovery trajectories. Climate change with changing temperature,
precipitation, and $CO_2$ concentration, etc. will also have an impact on the growth of individual trees and thus the
structure and composition of forests (e.g., Feng et al. 2018). The ED2-HuDi model developed in this study will be a
beneficial tool to understand the effects of frequent hurricane disturbances on forest recovery in the future under the
changing climate.

*Code and data availability.* The ED2-HuDi software is publicly available. The most up-to-date source code is available
at https://github.com/zhjiay5/ED2. The exact version used in this paper is archived on Zenodo
(https://dx.doi.org/10.5281/zenodo.5565063). Input data are available at https://doi.org/10.2737/RDS-2022-0025 and
https://doi.org/10.2737/RDS-2020-0012. Scripts to run the model and produce the plots for all the simulations
presented in this paper are also publicly available at http://www.hydrology.gatech.edu/.

*Author contributions.* R.L.B. conceptualized the work, T.H.S. provided field data and contributed ecological
interpretation of the results, R.L.B. and J.Z. developed the methodology and performed the analyses, J.Z. and M.L.
interpreted results, J.Z. wrote the first draft of the manuscript. All authors discussed results, and critically revised and
edited the manuscript.

*Competing interests.* Authors declare no competing interests.

*Acknowledgements.* We thank Paul Moorcroft, Xiangtao Xu, Elsa Ordway, Félicien Meunier and Erik Larson for
discussions on the model implementation and parameter sensitivity analyses. We acknowledge high-performance
computing support from Cheyenne (doi:10.5065/D6RX99HX) provided by NCAR's Computational and Information
Systems Laboratory, sponsored by the National Science Foundation. This work was supported by the National Science
Foundation (project EAR1331841) and K. Harrison Brown Family Chair. This research was supported in part by the
U.S. Department of Agriculture, Forest Service, and the USDA Forest Service International Institute of Tropical
Forestry works in collaboration with the University of Puerto Rico. The research was supported by the Jet Propulsion
Laboratory, California Institute of Technology, under a contract with the National Aeronautics and Space
Administration. M.L. was supported by the NASA Postdoctoral Program, administered by Universities Space
Research Association under contract with NASA, and by the Next Generation Ecosystem Experiments-Tropics,
funded by the U.S. Department of Energy, Office of Science, Office of Biological and Environmental Research. The
findings and conclusion in this publication are those of the authors and should not be construed to represent any official
USDA or U.S. government policy.

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
