# Peer review of "The Impact of Hurricane Disturbances on a Tropical Forest"

_Geoscientific Model Development, 2021_

## Author Comment (AC1)

This paper uses a forest dynamics model (ED2) to assess how hurricane disturbance effects composition and structure of a tropical forest. Key innovations in the work are (1) the development of a plant functional type specifically to represent palms, and (2) the implementation of hurricane disturbance in the ED2 model.

Overall, the paper is heavily focused on development of the model (warranted given the venue). However, I did think the biological context and implications could be more thoroughly presented, especially in the intro and discussion. I also felt there could be more reference to previous work on forest dynamics in the Luquillo Mountains (see detailed comments).

Although the manuscript is mainly well-written, there is room for improvement with respect to grammar / language. Please see detailed comments for some of the most important parts. There are numerous references on key points to papers that are in review. I do not know the policy of this journal but some things of the things being referenced are really critical to understand this paper properly (see detailed comments but, e.g., which species were classified into which PFTs, and on what basis?). I would think that some of these details should be included here in supplementary tables. At minimum, the papers in review should be posted to a pre-print server so the information cited is actually available.

Please note that many of the details on ED2 implementation / testing / calibration are outside of my expertise. Some of my detailed comments below do address these points, especially parts that I found could be more clear. Overall, however, I focus my comments on the general presentation of information in the paper and the links to the biology of the study system.

Thanks for the opportunity to review this manuscript, I hope the authors find my comments useful.

Re: We thank the reviewer for the insightful comments and suggestions. We have discussed previous studies on forest dynamics in the Luquillo Mountains in the Introduction. The biology of the ecosystem is beyond the scope of this manuscript, but we do have a separate manuscript on the way that focuses on links to biology of the ecosystem. Two out of the three references that were in review are now published and we have updated the citation; we have removed the one that is not published yet, and the information from that reference will be included in the supplementary of the manuscript. See the point-by-point response below.

INTRODUCTION

Comment #1 General: Overall, I think this section could be improved by making a more clear introduction to the fact that this study uses a vegetation model to understand hurricane effects on a tropical forest. As it stands, much of the text focuses on effects of hurricanes and then it jumps into PFTs and there is little/no text giving the context / background / rationale / etc. for using vegetation models, including using them to study hurricane effects. This is particularly surprising because there has been other work (including in wet forests of Puerto Rico) on modeling hurricane effects (see, e.g., Uriarte et al. 2009; Natural disturbance and human land use

as determinants of tropical forest dynamics: results from a forest simulator). Although not investigating hurricane impacts, another study using ED2 for wet forests in Puerto Rico did include palms as a 'late' PFT (Feng et al. 2017; Improving predictions of tropical forest response to climate change through integration of field studies and ecosystem modeling). It seems strange that this literature is not reviewed in the introduction.

Re: We agree with the reviewer that the introduction needed to contextualize the use of models. We have now reviewed relevant works and added the rationale for studying the effect of hurricanes on tropical forest using vegetation dynamics models.

"Although the immediate mortality and subsequent recovery of tropical forest from hurricane disturbances have been thoroughly studied via observations, the long-term effects of consecutive hurricane disturbances on tropical forests have rarely been studied. Models that can simulate the immediate mortality and subsequent recovery of an ecosystem can play a role in understanding potential mechanisms driving the mortality and recovery of the ecosystems and studying the long-term effects of disturbances, particularly if the nature of the disturbances are changing with climate. Uriarte et al. (2009) implemented hurricane disturbance in a forest simulator and investigated the long-term dynamics of forest composition, diversity, and structure. However, the biological and environmental processes of the forest simulator used are not dynamic and thus the model cannot simulate the adaptation of vegetation to the changes of environment (Jorgensen 2008). Vegetation dynamics models can account for changes in the ecosystem resulting from a changing environment (Medvigy et al. 2009; Longo et al. 2019b), and further allow us to explore scenarios via synthetic experiments and thus emulate what might happen in forests under novel environmental conditions. For example, Feng et al, (2018) used the Ecosystem Demography model (ED2) (Moorcroft et al. 2001) to study the impact of climate change on the forest studied in Uriarte et al. (2009). The ED2 model is a process-based vegetation dynamics model, it represents the size and age structure of the forest, and thus the model can represent the observed differential impact from disturbances (such as fire, drought, insects, land use change, and natural disturbances) across plants of different functional groups and size classes (e.g., Medvigy et al. 2012; Zhang et al. 2015; Miller et al. 2016; Trugman et al. 2016). However, the impacts of hurricane disturbances have not been implemented in vegetation dynamics models, and thus the long-term effects on the forest of a changing hurricane regime have not been investigated."

We have added a review on the Palm categorization in the Introduction.

"One important and distinct species in tropical forests in the Caribbean islands is the palm species *Prestoea montana*. Many studies in the Luquillo Mountains have either excluded palms from analysis (Zimmerman et al. 1994) or treat palms separately from other trees (Zimmerman et al. 1994; Uriarte et al. 2009), indeed they are monocots, not dicots like the other trees in the forest. A previous study that simulates the response of the forests in the Luquillo Mountains to climate change using the ED2 model categorized the palm species as a tree, Late PFT (Feng et al. 2018). But there are important differences, palms are more resistant to hurricane damage as compared to trees (Francis and Gillespie 1993; Uriarte et al. 2019) and are more resilient to hurricane disturbances due to their

high fecundity under open canopy (Lugo and Rivera Batlle 1987; Lugo et al. 1998) and high tolerance to shade (Ma et al. 2015). All those characteristics separate palms from other trees and favor the survival of palms after hurricane disturbances. We believe palms cannot be categorized into one of the existent PFT categories in the model and hence we define a new PFT—Palm.”

Comment #2 L 35-42: Exposure (to hurricane winds) seems to be an important missing factor here?

Re: Agree, we have added forest exposure to hurricane winds as an important factor in the contents.

“Hurricane-induced mortality varies with many factors, including hurricane severity (Parker et al. 2018), environmental conditions (Uriarte et al. 2019; Hall et al. 2020), **forest exposure to hurricane winds (Boose et al. 1994; Boose et al. 2004)**, forest structure (Zhang et al. 2022b), and traits and size of individual trees (Curran et al. 2008; Lewis and Bannar-Martin 2011). Trees with a larger diameter have been found to be more resistant to wind forces but more likely to suffer broken branches (Lewis and Bannar-Martin 2011). Species with higher wood density tend to suffer less from hurricane disturbances (Zimmerman et al. 1994; Curran et al. 2008). Hurricanes with heavier rainfall and stronger wind generally lead to higher mortality (Uriarte et al. 2019; Hall et al. 2020), **and forests that are more exposed to strong winds tend to have higher mortality (Uriarte et al. 2019)**. However, forests with a more wind-resistant structure and composition experience lower mortality even during a stronger hurricane event **or higher exposure** (Zhang et al. 2022b).”

Comment #3 L 45: It is not clear what is meant by "faster resprouting" - does it refer to sprouts being generated vs. time since disturbance? It seems to mean something different because the next sentence mentions time since disturbance as a separate point. Please clarify.

Re: “Faster resprouting” here means that sprouts are generated in a shorter time, referring to the timing of resprouts. “Faster resprouting” follows after “lower wood density”, meaning that the lower the wood density, the shorter the time to resprout. The next sentence says the resprouting rate or number of resprouts varies with time since disturbance, referring to the magnitude of resprouts. We have revised the sentences accordingly.

“Species with lower wood density have **shorter times to resprout** (Paz et al. 2018), higher growth rates (King et al. 2006), and shorter biomass recovery times (Curran et al. 2008). The **number of resprouts** of some species further varies with time since disturbance (Brokaw 1998; Zhang et al. in revision).”

Comment #4 L 47: Not clear what is meant by "higher recovery equilibrium". Equilibrium of what? Equilibrium in what sense?

Re: Revised it to “Less severe disturbances lead to a faster recovery and a higher level of stem density and aboveground biomass compared to the level observed prior to the disturbance”

Comment #5 L 51-54: It is a bit odd that this last part of the paragraph highlights a single article (Wang and Eltahir 2000) rather than providing some kind of summary / conclusion point about the preceding paragraph. If the Wang and Eltahir paper is very important to mention specifically then please provide more context.

Re: The summary of the paragraph, that forest recovery from hurricane disturbances depends on many factors, is in the beginning of the paragraph. Wang and Eltahir (2000) is an example of disturbance effects on ecosystem recovery, since this example is not exactly about hurricane disturbances we have removed this example.

Comment #6 L 55: The start to this paragraph is abrupt and could be more smoothly linked to the preceding text. I think the jump is mainly in the fact that all of a sudden you are talking about terrestrial biosphere models and PFTs but none of this has been introduced. A better link would help the overall flow here. The introduction is pretty short as it stands so there is space to develop this more.

Re: We added a paragraph to introduce the background and rationale for studying the effect of hurricanes on tropical forest using vegetation models, as in response to Comment #1. To link this paragraph with preceding contents, we revised the paragraph as

> "As mortality and recovery vary with species, the species composition of the forest is affected by hurricane disturbances. In modeling studies, it is impractical to incorporate individual species (tens or hundreds). To address variation in species diversity, there has been significant efforts in the past decades to incorporate functional diversity in vegetation dynamics models (Moorcroft et al. 2001; Sakschewski et al. 2016; Fisher et al. 2018; Fisher and Koven 2020). This effort acknowledges the variability in traits and trade-offs of species that exist in tropical forests (e.g., Baraloto et al. 2010)…."

Comment #7 L 57-63: More citations for the justification of the 3 PFTs mentioned would be useful.

Re: We have added more citations for the justification of the 3 PFTs in the manuscript:

> "... There has been a strong effort to incorporate functional diversity in vegetation dynamics models (Moorcroft et al. 2001; Sakschewski et al. 2016; Fisher et al. 2018; Fisher and Koven 2020). This effort acknowledges the variability in traits and trade-offs of species that exist in tropical forests (e.g., Baraloto et al. 2010). Three plant functional types (PFT) are identified for the species in tropical forests during a secondary succession after a disturbance; they are early, mid, and late successional PFTs (hereafter Early, Mid, and Late PFTs), corresponding to the three successional stages during the secondary succession (Kammesheidt 2000). Specifically, Early PFT dominates the early successional stage of the recovery, it includes fast growing pioneer species that have low wood density, establish and recruit in open gaps formed after disturbances and grow rapidly in the high light environment. Mid PFT dominates the mid successional stage after a disturbance, and includes species that have intermediate growth and are somewhat shade tolerant. Late PFT dominates the late successional stage and includes species that

have slow growth and are shade tolerant. Three PFTs is also a compromise between representing a range of life strategies while not adding too much complexity in model parameterizations (Moorcroft et al. 2001; Medlyn et al. 2005)"

Comment #8 L 63: I would advocate to start a new paragraph when introducing the palm. This is a focal point of the paper but it is kind of buried in this paragraph.

Re: We agree with the reviewer's suggestion and have updated the text accordingly.

Comment #9 L 64-65: Also reference: Uriarte, María, Jill Thompson, and Jess K. Zimmerman. 2019. "Hurricane María Tripled Stem Breaks and Doubled Tree Mortality Relative to Other Major Storms." Nature Communications 10 (1): 1362. https://www.nature.com/articles/s41467-019-09319-2

Re: Added as suggested.

Comment #10 L 69-70: "To account for these unique characteristics [in what? for what reason?], we define a Palm PFT." As other comments above, I think there needs to be an expansion of the introduction about vegetation modeling - building up to the overall aims of this study.

Re: We have revised as suggested, see the replies in Comment #1 for adding introduction about vegetation models and PFT categorization. Palms are more resistant to hurricane disturbances compared to trees due to lower mortality (Francis and Gillespie 1993; Uriarte et al. 2019) and more resilient to hurricane disturbances compared to trees due to their high fecundity under open canopy (Lugo and Rivera Batlle 1987; Lugo et al. 1998) and high tolerance to shade (Ma et al. 2015). All those characteristics separate palms from other trees and favor the survival of palms after hurricane disturbances.

Comment #11 L 74: Seems strange to say "The results indicate that a single hurricane disturbance has little impact on forest structure" when much of the introduction was spent discussing the various impacts hurricanes have on forest structure / composition. Does this really mean "long-term" forest structure? Please clarify.

Re: We were indeed referring to the long-term forest structure. We have revised as

"The results indicate that a scenario with a single hurricane disturbance has little **long-term** impact on forest structure and composition but enhances the aboveground biomass accumulation of a tropical rainforest - relative to **a no hurricane disturbance scenario**."

METHODS

Comment #12 L 92: "Since there is little knowledge about the traits of Palm." This is not a complete sentence and seems like it should be merged with the previous statement or otherwise revised.

Re: Merged with the previous sentence.

> "Yet, to date, none of the existing PFTs describe the traits of palms, even though palms are a globally abundant component of tropical forests (Muscarella et al. 2020)."

Comment #13 L 93: Not all palms have this low "wood density" and I think, in general, you should couch the statements about palms with something like "many palms" since it is such a diverse group and we do not actually know the degree to which these statements might be true or contradicted by some palms. In fact, the range of wood density for palms (Arecaceae) in the global wood density database (Chave et al. 2009, Zanne et al. 2009) is 0.180 - 0.883 (median = 0.54). (checked with the 'wdData' in the 'BIOMASS' R package, v. 2.1.5).

Re: We agree with the reviewer and we were referring to the palm species that occurs at our study site specifically. We revised the statement to

> "We know that **the palm species that occurs at our study site (*Prestoea montana*)** has a low wood density of 0.31 g cm$^{-3}$ (Lugo and Rivera Batlle 1987; Lugo et al. 1998; Swenson and Umana 2015)…".

Comment #14 L94-96: It is difficult to assess this decision because there are no details on the other traits used in the model. If palms "grow fast in open canopies like early tropical trees" then what is the reason to assume they "have the same probability distributions as those of late tropical trees"? I am guessing that wood density is strongly related to growth rate in high light conditions in the model. But isn't it also related to mortality rates (including in shade)? Since the introduction of this palm PFT is such a big part of this paper I think it should be explained in more detail here.

Re: The palm species at our study site has low "wood" density (Lugo and Rivera Batlle 1987; Lugo et al. 1998), which is like Early PFT. It is also tolerant to shade (Ma et al. 2015), which is like Late. Therefore, we assumed that the wood density of the Palm PFT has the same distribution as that of Early, and other physiological traits of Palm have the same distribution as those Late. Yes, both growth rate and mortality rates (aging and competition) are strongly related to the wood density in high light conditions in the model, and thus assuming Early-like wood density of Palm means Early-like growth rate and mortality of Palm. Palms would have a high growth rate in high light conditions like Early in the model, which is consistent with observations. The aging mortality for tropical PFTs has been parameterized as a function of wood density [$0.15 \times (1 - \rho_{PFT}/\rho_{LATE})$; Line 241], therefore the aging mortality of Palm in the model would be as high as Early, which is not consistent with observations, as we discussed in Section 2.3.2 Non-Hurricane Mortality. The palm species at our study site is observed to have a very low aging mortality, disproportionate to its wood density, and thus we assumed that the aging mortality of Palm is independent of its wood density, and that it is the same as Late.

Comment #15 L 104-105: Palms can be shorter than other trees, given the same DBH but I am a bit skeptical of these allometric relationships. For one thing, when I plot them, palms can never reach more than about 13 m height at the maximum diameter (20 cm), which is too short but other trees are predicted to reach unreasonably tall heights for this forest (~60 m for early PFT at the maximum 90 cm diameter). The only justification for these fitted parameters is from a paper by the authors 'in revision'.

Re: Based on field measurements at the site, palms are indeed shorter and barely exceed 18 m height and 20 cm DBH. The figure below (part of the manuscript in review) provides the H-DBH relationships from the census observations at BEW. The maximum DBH is less than 30 cm for Early and less than 70 cm for Mid and Late at our study site, and the maximum height of trees is around 25 m for Early and 30-35 m for Mid and Late. We will add this figure as a supplementary if the paper in revision is still unpublished.

[Figure]

Figure AC1: The Height and DBH pairs at BEW for each PFT. The gray dots are observations, and the blue line is the fitted relationship between H and DBH. The $R^2$ and P-value for the relationship are shown in the figure.

Comment #16 L 107: This sentence requires a citation.

Re: Citation added.

Comment #17 L 108: This sentence needs revision for clarity / grammar.

Re: Revised as "A previous study implementing liana to the ED2 model also experienced similar issues (di Porcia e Brugnera et al. 2019). They used an allometry for lianas with DBH between 3 and 20 cm and then for lianas with DBH less than 3 cm they used the allometry of early successional trees".

Comment #18 L 112-114: I am questioning the ramifications of these 'tricks' implemented in the model to help allow palms to survive despite their allometry.

Re: As we observed from the census observations, palms will barely exceed 18 m in height, and thus we set a maximum height threshold for Palm as 18 m. When palms stop growing in height it does not mean that they will die or stop accumulating carbon. As palms are shade tolerant (like Late PFT), they can still photosynthesize and accumulate carbon, they allocate the accumulated carbon to survival (making new fronds) and reproduction instead of growth. This is consistent with the height and DBH observations at our study site. See the figure in response to Comment #15.

Comment #19 L 115-117: Here the authors use default allometry of Early PFT for Palms but this seems inconsistent with the statement in L 94-96 about "...we assume that the traits of Palm have the same probability distributions as those of late tropical trees..." Please clarify.

Re: Structural traits (allometries) and physiological traits are separated. We have revised it as

> "…we assume that the **physiological** traits of Palm have the same probability distributions as those of late tropical trees except for wood density which is assumed the same as that of early tropical trees. **The allometries of Palm are discussed separately in the next section.**"

Comment #20 L 124: A bit confused by "(sc) is the ratio of the cohort density that survived to the cohort density *before* the disturbance,"... should this not be the proportion that survives after the disturbance? Since sc=1-λc (L 127), and that λc "varies with hurricane strength, ...", it makes sense that sc would be post-hurricane survival... Please clarify / revise.

Re: The reviewer is correct, "sc" is the ratio of the density after the hurricane to the density before the disturbance. We revised the text as:

> "The survivorship of each cohort (sc) is the ratio of the cohort density that survived **after the disturbance** to the cohort density before the disturbance"

Comment #21 L 135: It seems like "Given mortality, the rate of each cohort (λc)" should be revised to "Given the mortality rate of each cohort (λc)"?

Re: Here "Given the mortality" means if the mortality occurs, since mortality is a binary function of wind speed (if wind speed is less than a threshold, mortality will not occur). We revised it to

> "**If mortality occurs (i.e., wind speed exceeds the threshold)**, the mortality rate of each cohort (λc)…".

Comment #22 L 138-145: It is really not clear what is showing on the x-axis of these figures (x: proportion of large stems). How can this be the same during a given hurricane event for all PFTs? And why is mortality lower for all PFTs from hurricane Maria compared to Hugo? This section needs clarification.

Re: The x axes are the proportion of large stems (DBH ≥ 10 cm) of the forest. For example, if we have 1000 stems in the forest and 500 of them are large stems with DBH ≥ 10 cm, then the

proportion of large stems of the forest is 500/1000=0.5. It describes the size structure of the forest. Thus, it is a function of only the size structure of the forest, no matter what the PFT composition is. The mortality is lower for all PFTs from hurricane Maria compared to hurricane Hugo because the forest at BEW was less exposed to hurricane Maria than to hurricane Hugo, and the forest at BEW had higher abundance of large stems and palms, which made the stand more wind-resistant during hurricane Maria. Those comparisons are discussed in detail in Zhang et al. (2022b). We have revised the figure legend to explain the proportion of large stems.

> **Figure 1. The mortality as a function of the size structure of the forest for each PFT and DBH class. The size structure is represented as the proportion of large stems (DBH ≥ 10 cm) to the total number of stems in the forest (DBH ≥ 2.5 cm).** The dots represent observed mortality and proportion of large stems pairs from hurricane Hugo and hurricane Maria (Zhang et al. 2022b). Four colors represent four PFTs. The solid lines represent the estimated mortality as a logistic function of the proportion of large stems. The panel on the left is for small stems and that on the right is for large stems."

Comment #23 L 146: Again some very relevant references seem to be missing. In particular Uriarte et al. 2009 (Natural disturbance and human land use as determinants of tropical forest dynamics: results from a forest simulator) and Uriarte et al. 2012 (Multidimensional trade-offs in species responses to disturbance: implications for diversity in a subtropical forest).

Re: We included the references as suggested by the reviewer.

Comment #24 L 150-160: Please clarify the data upon which these functions are based.

Re: These functions are based on the recruitment with time since hurricane Hugo, and the data are from Zhang et al. (2022a). The reference is now appropriately cited.

Comment #25 L 170: I am missing details on the basis by which species were assigned to PFTS... this seems too important to have only as cited in a work 'in review'.

Re: The reference that was in review is now published, and the PFT of each species is presented in Table S1 in Zhang et al. (2022b). We now add a brief explanation of the criteria used to assign species into PFTs.

> "Following Zhang et al. (2022b), species are categorized into four PFTs **according to their successional status based on previous studies** (Walker 1991; Schoealter and Ganio 1999; Uriarte et al. 2005; Muscarella et al. 2013; Heartsill Scalley 2017; Feng et al. 2018): early, mid, late successional tropical trees, and palms (Early, Mid, Late, and Palm PFT, respectively)."

Comment #26 L 161-173: It seems that perhaps this description of the census data should go earlier in the text? But more importantly, it is a bit problematic to have such important references to work that is 'in review' (not to mention there are two Zhang et al. 'in review' papers so we don't know for sure which one is being cited here). Perhaps now this paper is

published?  What is the plan if this manuscript is accepted before the outcome of the one 'in review'?  It seems like posting a pre-print of the other work could be at least a partial solution.

Re: Both the data (Zhang et al. 2022a) and the paper (Zhang et al. 2022b) are published now, and we updated the references accordingly. We exchanged the order of Section 2.2 Census Observations (now Section 2.1) and Section 2.1 Model Description (now Section 2.2).

Comment #27 L 172: Does the Scatena et al. (1993) biomass allometry apply to palms?  Should a caveat be included here?

Re: The AGB-DBH relationship from Scatena et al. (1993) applies to all dicot trees. Scatena et al. (1993) did provide the AGB-Height(H) relationship for Palm. Since we have the H-DBH relationship estimated from our census data (Section 2.1.2 Modifying the Allometric Relationship, and also see the response to comment #15), we can estimate the AGB-DBH relationship from AGB-H and H-DBH relationships. Now we have updated the AGB of Palm by using the AGB-H relationship from Scatena et al. (1993) and the H-DBH relationship from our census data. Since Palm AGB accounts for less than 10% of the total AGB of the forest (palms have smaller DBH and Height than most trees in the forest), the difference of total AGB (including all PFTs) between estimates using AGB-DBH for Palm and estimates using AGB-H for Palm is very small (<0.2%). We have updated the texts and corresponding figures (Figure 4, Figure S3, Figure S4, and Figure S8).

> "The observed aboveground biomass (AGB) of Early, Mid, and Late PFTs are estimated from DBH using the AGB-DBH relationship from Scatena et al. (1993); the AGB of Palm PFT is estimated from the AGB-Height relationship of *P. montana* from Scatena et al. (1993) and the Height-DBH relationship of Palm PFT from the census observations at our study site (see Section 2.1.2 Modifying the Allometric Relationship)."

Comment #28 L 193-194: Clarify specifically the reason why the dark respiration factor from Feng et al. 2018 has "too wide a range".  I am not familiar with the paper cited, which seems relevant but from a completely different study system.

Re: We agree that Wang et al. (2013) is from a different study system, but we consulted with Paul Moorcroft and Xiangtao Xu, the developers of the ED2 model, and they suggested that 0.005–0.0175 would be more appropriate. We have revised the text as

> "From a different study system, Wang et al. (2013) constrained the dark respiration factor from 0.01–0.03 to 0.01–0.016 by assimilating observations of model output variables. Following Wang et al. (2013), we restrict the dark respiration factor to a smaller range with a uniform distribution between 0.005 and 0.0175 for each PFT."

Comment #29 L 197: Please clarify: you say, "clumping factor is defined as the projected area of leaves per unit ground area" but then the following details about ranging from 0-1 is more about the relative clumping of leaves over a given unit area

Re: Clumping factor is the ratio of effective LAI to the total LAI. Revised accordingly.

Comment #30 L 237: Why do background mortality rate for large stems? Treefall disturbance rate between small and large stems seems very similar - what data is this based on?

Re: We revised the text to clarify the points raised by the reviewer:

> "Background mortality rate is zero for large stems because, following Moorcroft et al. (2001), this mortality is accounted for in the treefall disturbance rate (i.e., the background mortality of large trees is what causes the treefall disturbance). The treefall disturbance rate mortality is a combination of the area impacted by treefall disturbance and the survivorship of this disturbance. By default, in ED2 it is assumed that the treefall disturbance rate is 0.014 $yr^{-1}$, survivorship of large trees to treefall disturbance is zero (and thus overall treefall mortality is 0.014 $yr^{-1}$), and survivorship of small trees to treefall disturbance is 10% (and thus overall treefall mortality is 0.0126 $yr^{-1}$)."

Comment #31 L 250-260: I find this part on palm recruitment to be confusing, in part, because there seems to be a disconnect between the model and data/biology of the system regarding palm "seedlings" and recruitment. Indeed, these palms produce abundant seeds and seedling density could perhaps be considered similar to early successional species. However, the decay of palm seedling abundance with time since disturbance is less dramatic than for early successional species, to my knowledge (there is relevant data available on this at least from the LFDP and prior studies). But how does this really relate to the 'observed recruitment of palms' in the data? The text is not clear about what "recruitment" of palms actually means in the data (at what height / diameter do they enter the census?). These are typically not at all "seedlings" since palms produce robust diameter stems prior to growing taller. If the model considers newly recruited individuals as those represented in the data, then it may not really be reasonable to assume that the "seedling" density of palms is similar to that of the early successional PFT. I think some additional details and work revising this section would be valuable.

Re: The seedling density in this context means the seedling density produced each year after the disturbance, it does not mean the accumulated seedling density over the years. Therefore, the decreasing density with time does not mean the decaying of seedling through time but means that the seedlings produced each year decreases with time (the accumulated seedling still increases with time). The recruitment refers to stems entering the census (DBH≥2.5 cm and H≥1.5 m) each year. If there were more seedlings in year 1 than year 10 after a disturbance, then it is reasonable to assume that there were more recruits in year 1 than year 10, and vice versa. As we do not have seedlings but only recruited stems in our census data, we assumed that seedling density has the same response (varying with time since disturbance) as recruitment, but not necessarily the same magnitude (density) as recruitment. We did not assume that palm has the same seedling as Early but assumed that palm has the same response (decreasing with time since disturbance) as Early, and the parameters related to the decrease are different for Early and Palm. We have revised the text to clarify our assumptions

> "..., we compared the recovery density schemes with the observed recruitment of Palms **(stems entering the census with DBH≥2.5 cm and H≥1.5 m each year). As we do not have seedlings but only recruited stems in our census data, we assumed that seedling density has the same response (varying with time since disturbance) as recruitment,**

**but not necessarily the same magnitude (density) as recruitment. Based on the census data,** there were 37, 64, 50, 34, and 32 palms recruited in the 85 plots (78.5 m$^2$ each plot) in 1994, 1999, 2004, 2009, and 2014 censuses, respectively, which corresponds to 0.0011, 0.0019, 0.0015, 0.0010, and 0.0010 individuals m$^{-2}$ yr$^{-1}$ after 5, 10, 15, 20, and 25 years of the Hugo disturbance. In other words, the recruitment decreases to half of the starting level in 20–25 years, or a decaying factor $\alpha \approx 0.03$ yr$^{-1}$. We assume that the seedling density has the same decaying rate as the recruitment density and thus we select the seedling density scheme $n_0$=0.02 individuals m$^{-2}$ yr$^{-1}$ and $\alpha$=0.03 yr$^{-1}$ as the seedling recovery scheme for Palm."

Comment #32 L 314-317: It is somewhat difficult to assess this with knowledge of the study system without knowing which species are included in each PFT. I am missing a table showing this. L 170 says this information is in Zhang et al. (in review) but seems to important to simple be cited in another paper, especially when that paper is not published yet.

Re: The paper has been published now (Zhang et al. 2022b). The table is in the supplementary information of Zhang et al. (2022b).

Comment #33 L 319: Is 25% underestimation and 38% overestimation considered "consistent with observations"?

Re: We have revised it as "The simulated stem density of Late and Palm are also within one standard deviation of the observations although the model predictions suggest 25% underestimation and 38% overestimation, respectively".

RESULTS

Comment #34 L 327: It would be good to include units and more informative labels on the figure itself. The legend does not seem to define the red line, which it should.

Re: Following the reviewer's suggestion, we added units and labels in the figure now, and revised both the figure and the legend to define the red line (simulated time series)

[Figure]

"**Figure 3.** Time series of variables from observation (**dots and error bars**) and the optimal simulation (**red lines**). (a)-(d) stem density of all trees (n; DBH $\geq$ 2.5 cm) (individuals m$^{-2}$) for Early, Mid, Late, and Palm PFTs, respectively. (e)-(h) diameter growth rate (GR; cm (5yrs)$^{-1}$) for the four PFTs; (i)-(l) basal area (BA; cm$^2$ m$^{-2}$) for the four PFTs. The dots and the error bars represent the means and the one standard deviations from the means across the 85 plots. Period between 1989–2014 is for model calibration and period between 2015–2018 is for model validation (shaded)."

Comment #35 L 350: Instead of referring to wood density of Prestoea decurrens, the authors could cite measurements of wood density for the study species directly (0.31 g cm^3), which is available here: https://datadryad.org/stash/dataset/doi:10.5061/dryad.j2r53

Re: Revised as suggested.

Comment #36 L 359-362: Is it also possible that the posterior PDFs do not change much from the priors because of some characteristics / amount of data going into the models? Attributing this fact to some reason seems like more of a discussion point than a result.

Re: The sentence has been revised so that it does not sound like a discussion.

"The posterior PDFs of some parameters (i.e., carboxylation rate, specific leaf area, leaf width, stomatal slope, and wood density), which are well constrained by observational trait data (Feng et al. 2018), do not change much from the priors (the maximum difference between the prior and posterior CDFs is generally less than 0.1)."

Comment #37 L 449: 2 cm yr-1 increment in DBH is extremely high and I don't know where this number comes from? The abstract for the paper cited (Brandeis 2009) says, "...growth rate averaged... 0.36 cm/year in subtropical wet/rain forests, and 0.20 cm/year in lower montane forests." (https://www.fs.usda.gov/treesearch/pubs/34208). This value is also more than 2x higher than the maximum DBH growth rate shown in Figure 10. Something here seems to need clarification.

Re: Table 1 in Brandeis (2009) lists the mean (Mean), standard error of the mean (SE), standard deviation of the mean (SD), and maximum (Max) of observed DBH periodic annual increments and the Results section says "Individual tree of several species exhibited growth rates over 2.5 cm/year in this study. They were … *Cecropia schreberiana* Mid. (4.30 cm/year), …, *Inga laurina* (Sw.) Willd. (3.32 cm/year), … *Guarea guidonia* (L.) Sleumer (2.90 cm/year), *Inga vera* Willd. (2.88 cm/year), ..." We used this 2 cm/year DBH increment to explain that a 180-cm-DBH in the model simulation is possible (a maximum of 89 cm DBH was observed in the 2017 census).

We acknowledge that 2 cm/year DBH increment is indeed extremely high. Now, instead of persuading that 200 cm DBH is possible, we have revised the text to acknowledge that the model has overestimation on DBH.

> "The maximum DBH is far larger than that we observed (89 cm in 2017), which could be an overestimation due to no nutrient limitation."

Comment #38 L 455: Change "the ones" to "the experiments"

Re: Revised as suggested.

Comment #39 L 465-467: The growth rates shown here for late PFT trees are quite a bit higher than what is typical in these forests (see comment about L 449).

Re: The growth rate for Late PFT is around 0.4 cm/yr for the experiment IhugoHn (no hurricane disturbance; Figure 10), which is similar to the mean growth rate of subtropical moist forests (0.37 cm/yr) and subtropical wet/rain forests (0.36 cm/yr) in Brandeis (2009). The growth rates for experiments with hurricane disturbances (IhugoH1 and ImariaH1) (~0.5-0.8 cm/yr) are quite a bit higher than that of IhugoHn (or the typical values in these forests), which is a result of the disturbance: opening the canopy and enhancing the growth of trees in the gaps. As we can see from observations (Figure 3), the growth rate of Late decreases with time since disturbance (although not as significant as other PFTs), evidencing the effect of hurricane disturbance on growth rate. Moreover, we compared the simulated growth rate with observations in Figure 3, and the simulated values are within the one standard deviation of the observed means at our study site.

DISCUSSION

Comment #40 In general, this section is very short and it feels like there is a lot of work to be done in terms of putting the pieces together for a robust interpretation of the study results. Also the discussion focuses almost entirely on the modeling exercise but extremely little points back to the biology of the system.

Re: This is a Development and Technical Paper, and thus we focused heavily on the modeling development aspect, the biology of the system is beyond the scope of this manuscript. In fact, another manuscript of ours that focuses on links to biology of the ecosystem is on the way.

Comment #41 L 486: I would like to see a brief introduction to the discussion section that quickly summarizes the key findings and provides a structure to what we can expect to read in the rest of the section.

Re: We added a brief introduction to the discussion section.

> "We developed a hurricane module (including a mortality module and a recovery module) for the ED2-HuDi model, based on census observations. We then applied a parameter estimation algorithm, GLUE, to calibrate important parameters in the model and selected the optimal parameter set for the final model simulation. However, because the observations are limited to only two hurricane events, the hurricane module may be biased toward the two observations. The simulation results show some discrepancies with observations, and these discrepancies could be in part due to the GLUE approach and parameter uncertainties. Here we discuss the uncertainty associated with the developed hurricane module, the limitations and advantages of the GLUE framework, and the uncertainties of model outputs."

Comment #42 L 510-517: RE: clumping factor: It is not clear what are the implications for the clumping factor. The value controls LAI but what does that mean for the simulated dynamics?

Re: As described in the response to Comment #29, clumping factor is the effective LAI divided by the total LAI. Lower clumping factor means lower effective LAI, which will allow more light to reach the ground and thus increase the photosynthesis of the understory plants.

Comment #43 L 529: "...vegetation dynamic**s**."

Re: Revised as suggested.

**Anonymous Referee #2**

This study examined the impact of hurricane disturbances on tropical forests by use of modeling. Therefore, authors extended the ED2 model by a new disturbance component on hurricanes and an additional PFT of palms, and calibrated the model with the GLUE approach for a forest site in Puerto Rico. With a sensitivity and scenario analysis, authors discussed the uncertainty of their model calibration and demonstrated the impact of forest state and structure before a hurricane event on the recovery of forests. The study is comprehensively conducted and described and the manuscript clearly written. I have a few points to recommend for minor improvement of the manuscript.

Re: We are glad that the reviewer thought our study is comprehensively conducted and described and the manuscript is clearly written.

General comments:

The relevance of studying hurricane impacts on tropical forests and why modelling is an important tool besides observations should be emphasized more clearly in the introduction, abstract and conclusion. What is your motivation of extending the ED2 model by hurricane disturbances? E.g. in the conclusion (page 20, lines 528-529) you state that no model has implemented hurricane disturbances so far. Please write in more detail about the relevance of such applications. Which benefits can models provide in this context (besides observations)? Few points are mentioned in the conclusion, but the relevance of your study should also be emphasized in the abstract and introduction.

Further, why did you choose the ED2 model? How is it related to other models studying disturbance impacts on tropical forests in general. Please shortly relate your work to the current scientific literature on modeling tropical forests and disturbances in general.

Re: Modeling is an important tool besides observations because the observations are limited to immediate damage and short- to intermediate-term recovery (months to tens of years) from disturbances. By implementing hurricane disturbances modules in vegetation dynamics models, we can, for the first time, study the long-term effects of hurricane disturbances on tropical forest structure and composition. Due to climate change, hurricanes are projected to increase and reach subtropical areas, which will subject tropical forests to disturbances with higher frequency and intensity. Process-based vegetation dynamics models implemented with hurricane disturbance allow us to understand potential mechanisms driving the mortality and recovery of the ecosystems and to explore what might happen under novel environmental conditions. The ED2 model represents the size and age structure of the forest, and thus the model can represent the differential impact of disturbances across plants of different functional groups and size classes, and thus allows us to study the impact on forest structure and PFT composition. We added a section in the Introduction to review the literature on modeling tropical forests and disturbances in general, as replied to Reviewer #1 Comment #1.

Minor points:

1) page 2, line 31: Can you provide few numbers? How often do they occur on average?

Re: Hurricane frequency varies with locations. For our study site, the Luquillo Experimental Forest at Puerto Rico, it has been estimated that hurricanes will pass directly over the LEF once every 62 years and within 60 km once every 22 years (Scatena 1989). We have now provided a few examples on hurricane damages to trees and aboveground biomass:

> "For example, hurricane Hugo in 1989 uprooted and snapped 20% of the trees at El Verde in the Luquillo Experimental Forest (LEF), Puerto Rico (Walker 1991; Walker et al. 1992; Zimmerman et al. 1994) and reduced the aboveground biomass by 50% at Bisley in the LEF (Scatena et al. 1993; Heartsill Scalley et al. 2010). Hurricane Katrina in 2005 damaged about 320 million large trees on U.S. Gulf Coast forests, and the damaged trees are equivalent to 50-140% of the net annual U.S. carbon sink (Chambers et al. 2007)."

2) section 2.1: I think the general model description of ED2 could benefit from a summarized description of its basic structure. Although you refer to literature references, it is important to have general information on the main processes (recruitment, growth, competition, mortality) also in your manuscript. Especially in section 2.3.2 you mention different mortality sources in the absence of hurricane disturbances and to understand this, already more information in the section on the model description is required.

Re: We have added description on the main processes of the ED2 model.

> "A cohort accumulates carbon through photosynthesis, and the net accumulated carbon (i.e., gross primary productivity minus respiration and maintenance of living tissues) will be used for growth and reproduction. When a cohort is mature, reaching the maturity reproductive height (e.g., 18 m), the cohort will allocate a portion of carbon to reproduction (e.g., 30% of net carbon accumulation to seeds, flowers, and fruits), and the rest of the net accumulated carbon will be used for structural growth. Structural growth is quantified by the increase of DBH through structural biomass-DBH allometries; stem height, leaf biomass, and crown area are then scaled given the H-DBH, leaf biomass-DBH, and crown-DBH allometries. Each cohort will also experience mortality from multiple factors, including aging, competition, and disturbance, which will be described in detail in Section 2.3.2."

3) page 5 line 122: Is lambda_d affecting the entire patch or a fraction of a patch?

Re: "For any patch with pre-disturbance area $A$, the area that is affected by disturbance ($A_d$) is proportional to $\lambda_d$, following Moorcroft et al. (2001): $A_d = A\,[1 - \exp(-\lambda_d \Delta t)]$. The disturbed area ($A_d$) will be disturbed and become a new patch (age 0), and the population within the new patch will be determined by the survivorship to disturbance. The remaining area ($A-A_d$) will remain undisturbed, and the stem density will remain unchanged." We included this information in the text.

4) page 6 line 165: Please add the size of a plot. You mention it later in the manuscript (page 9, line 261), but it should already appear here.

Re: Added "Each plot is a 10-meter diameter circle and plots are 40 meters apart extending 13 hectares."

5) page 9 line 247: How would this assumption affect your model simulations of long-term studies (e.g. longer than 100 years)?

Re: If the aging mortality of Palm were set to the default (~0.1 yr$^{-1}$), then the overall mortality of Palm (including aging, competition, and disturbances) would be greater than 0.1 yr$^{-1}$, which is much higher than the observed Palm mortality at our study site (~0.03 yr$^{-1}$; Figure S5f and g). With this 0-aging mortality assumption, the mortality of palms come from only disturbances and competition (~0.03 yr$^{-1}$; Figure S5f and g), which is consistent with the mortality observed at our study site (0.03 yr$^{-1}$ on average; Figure S5a).

6) page 17, Fig. 9f: The stem proportion of Mid PFTs seems to still decline (if simulating longer than 112 years; similarly basal area, Fig. 9c) in some scenarios. Nevertheless, you state that the forest reaches a steady state after 80 years (page 18, line 439). How did you determine its steady state? Further, do you have an explanation why Mid PFTs are still declining (in comparison to the other PFTs) and "mostly have small stems" (page 18, line 443-444)?

Re: Forests are dynamic systems, and their state will always change. The steady state of a variable is obtained when the 30-year moving average of the variable has less than 1% change compared to the previous year. The reviewer is correct in that the stem proportion of Mid still declines after 80 years, however the decline is substantially reduced at that point, and the rate of change based on the 30-year moving average is less than 1% (see the figure below).

[Figure]

Figure AC2: Same as Fig. 9, except for the percent change of the 30-year moving average of the variables. The black dashed lines are the −1% and 1% thresholds.

The Mid PFT declines because it is less competitive than other PFTs due to its parameter values, such as higher dark and growth respiration rates (Table 1). Most of the Late stems are small

DBH stems possibly because a higher portion of carbon was allocated to roots (higher fine root allocation) instead of growth compared to other PFTs.

Specific comments:

1) page 3, lines 51-54: Difficult to understand in relation to the previous sentences. Can you rephrase? (e.g. what is "the initial vegetation condition"?)

Re: "the initial vegetation condition" means the vegetation condition right before a disturbance. The two sentences have been removed since this example is not about forest recovery from hurricane disturbances (the main point of this paragraph) (see also replies to Reviewer #1 Comment #5).

2) check some spelling and grammar in your manuscript, e.g. page 4 (line 92, "Since …"), page 4 (line 108, "They then were to use …"), page 10 (line 285-286), page 10 (line 300, "of" is missing after "impact"), page 12 (line 341, "compated"), page 19 (line 479, "AB"), page 19 (line 499, "utlized"), page 20 (line 518)

Re: The spelling and grammar issues have been corrected.

3) define and describe variables the first time you mention them in the manuscript (e.g. page 4, line 98, H and DBH should be defined including their units)

Re: Revised as suggested.

**Anonymous Referee #3**

Summary

Cyclonic storms are one of the major natural disturbances in tropical forests, and the intensity of tropical cyclones has been projected to increase over this century. Characterizing hurricane damage and post-hurricane recovery is critical for estimating forest resilience and the fate of tropical forests. This study implements a new hurricane module in a dynamic vegetation model, the Ecosystem Demography model (ED), to account for hurricane-caused tree mortality and post-hurricane recovery, which is primarily driven by wind speed, forest structure, and functional diversity. The study also added a new plant functional type for Palms, which can differ from other dicot tropical tree species in terms of ecophysiology and responses to hurricanes. The study performs some model sensitivity tests using GLUE and provides much detailed information on the methodology and results. Altogether, the study highlights the importance of representing the hurricane effect in terrestrial biosphere models.

Comments:

The manuscript provides a comprehensive model calibration and sensitivity analysis within the framework of GLUE. The materials and methodology are clear. Major comments are listed below.

Re: We are glad that the reviewer thought our materials and methodology are clear. The point-by-point replies are listed below.

Comment #1 First, the hurricane module is way less discussed in the study compared with functional diversity, and the Palm PFT despite the title focusing more on hurricanes. The method section describes a general framework to include hurricane module (i.e. link hurricane damage to hurricane intensity, forest structure, and species diversity). However, it is not clear what is the uncertainty/biases associated with the framework, which I believe can be large. For example,

- the key relationship in the hurricane module is parameterized by only two points (Fig. 1) and the low hurricane mortality for early successional big trees (Fig. 1b) is somewhat suspicious when the large tree fraction is small.
- Shouldn't Palms have generally lower mortality compared with other PFTs under hurricanes?
- It is also mentioned that partial crown damage is prevalent under hurricanes, which is not included in this framework and not even discussed.
- What are the key hurricane-related parameters that make the model capture changes in stem density and composition? (Fig. 4)

Given the title, readers would expect some in-depth exploration/discussion of the hurricane module and parameterization. Therefore, I would recommend including more sensitivity tests for the hurricane module or changing the title and intro to focus on Palm PFT.

Re: The hurricane module was explained in detail in the Method section Section 2.1.3 (now Section 2.2.3). The uncertainty of the recovery module was discussed briefly for the Palm PFT in Section 2.3.2. We have added a discussion about the uncertainty of the hurricane module in the Discussion section.

- That only two points are used in parameterizing the relationship of the hurricane module is because we only have two observations (Zhang et al. 2022b). The suspicious low mortality for large Early PFT when large stem proportion is low was due to the fitting. The parameterizing of the hurricane module could be improved given more observations.
- In the current parameterization, palms should have lower mortality than other PFTs, and Fig. 1 shows that, except for the low mortality of large stems of Early PFT.
- Crown damage is a part of hurricane damage to the forest, but the census observations used to calibrate the model do not have crown damage information, therefore we did not include crown damage here. We added a brief discussion.
- The key parameters that capture changes in stem density and composition are disturbance rate of forest area ($\lambda_d$) and survivorship of each cohort ($s_c$) from the mortality module, and initial seedling density ($n_s$) and decay factor of seedling density ($\alpha$) with time since disturbance from the recovery module (Section 2.2.3). $\lambda_d$ applies on patch area, and $s_c$, $n_s$, and $\alpha$ apply on stem density and are PFT dependent, therefore, the last three parameters explicitly capture the changes in stem density and PFT composition during the immediate mortality and the subsequent recovery processes.

We add a discussion section for the uncertainty of the hurricane module (section 4.1).

"4.1    Uncertainty of the hurricane module

We included a hurricane mortality module and a hurricane recovery module for hurricane disturbance. Crown damage is also an important part of hurricane disturbance and could have an important impact on forest structure and carbon accumulation (Leitold et al. 2021), but we did not include crown damage in the hurricane disturbance module because the census data used to develop and calibrate the module do not include crown damage information. The hurricane mortality module was developed based on observations from two hurricane events at the study site. The relationship between mortality and forest size structure (proportion of large stems) was fitted to a logistic function (Figure 1) for each PFT and DBH class. Generally, Palm PFT has a lower mortality than other PFTs, but Palm mortality was higher (11% for Palm, 9% for Mid, and 3% for Late) when the forest was dominated by large stems (e.g., large stem proportion is 0.6), except for the high mortality of 39% for Early (Figure 1b). This was due to the high mortality of Palm during Maria, which was a result of diseases (Zhang et al. 2022b; Heartsill Scalley 2017). The mortality of large-stem Early PFT is significantly different from other PFTs, this difference was due to the significantly higher mortality of large stem Early during hurricane Maria compared to other PFTs. Such high mortality of large stems Early may be a result of other factors besides hurricane disturbance, and it could be further studied if there were more observations. In the future, observations from other study sites could be used to improve the hurricane disturbance module.

There are four critical parameters associated with the hurricane disturbance module, including disturbance rate of forest area ($\lambda_d$) and survivorship of each cohort ($s_c$) from the mortality module, initial seedling density ($n_s$) and decay factor of seedling density with time since disturbance ($\alpha$) from the recovery module. We tested the sensitivity of the parameters of the recovery module but did not test the uncertainty of the parameters of the mortality module because the values are from observations at the study site. For future studies using this module, either testing the uncertainty of the parameters or using site specific values are encouraged."

Comment #2 Second, the GLUE trait optimization seems to be quite sensitive to light-related parameters. For example, the equilibrium clumping factor has a rather low value (< 0.4 while reported values are >0.6 over tropical forests). Quantum efficiency and dark respiration are dominating the variance (Fig.8). I think this might be because the canopy structure and light environment of the model are highly biased. Fig.S2 shows the initial LAI can exceed 8 (constrained by observed demography I guess?), which is rather high. This might explain why optimal Clf is so low and can be caused by biases in allometry (in fact, the allometric parameters can have huge effects but are not tested in the study). Meanwhile, this model does not consider acclimation to understory light. It is understandable that fully addressing these issues is challenging but they need to be acknowledged and discussed.

Re: We agree that the GLUE trait optimization is quite sensitive to light-related parameters and that allometric parameters may affect the optimization. We will add discussions about the mentioned aspects.

"The optimization is sensitive to light-related parameters, such as clumping factor, quantum efficiency, and dark respiration (Fig. 8). This is consistent with Meunier et al. (2021) who found that light limitation contributes partly to model uncertainties. The clumping factor we calibrated for our study site is lower than that from other locations (He et al. 2012), which could be due to uncertainties of the allometries and estimates on leaf area index. As discussed in Shiklomanov et al. (2021), the ED2 model has a less robust estimation on LAI because of structural errors in representing direct radiation backscatter. Acclimation to understory light is not considered in this model, however, traits respond strongly to light environments (Lloyd et al. 2010; Keenan. and Niinemets 2016), therefore it needs to be considered in future developments (Xu and Trugman 2021)."

Comment #3 Third, the hurricane impact and recovery simulations are interesting but are underexplored. Why only look at the impacts on equilibrium forest structure? Shouldn't the time scale be the average return interval of hurricanes in Puerto Rico? What about using additional initial conditions by sub-sampling different plots?

Re: This paper is a Development and Technical Paper, so we focused on the development of the model. The impact and recovery simulations of frequent hurricane disturbances are presented in a separate paper, hopefully to be submitted soon.

Minor comments:

Comment #4 Line 55: the transition from hurricane impact (the previous paragraph) to functional diversity/PFT (this paragraph) seems somewhat abrupt. Some elaboration about why palm is unique, or why we need to incorporate this particular PFT in the context of hurricane disturbance will be helpful, e.g., the relative abundance of palm in hurricane-prone sites. And this information about palm should probably come before the explanation about early and late-successional species (line 58).

Re: This transition issue has been pointed out by other two reviewers as well. We will revise the introduction by adding the background and rationale for studying the hurricane impacts on tropical forests using vegetation dynamics models. Hurricane impacts vary with species, but it is impractical to incorporate each and individual species in modeling studies, and therefore PFT categories are incorporated. Palms are different from trees, as palms are more resistant to hurricane disturbances (lower mortality) and more resilient to hurricane disturbances (high fecundity and high tolerance to shade) compared to other trees. Those characteristics would favor the survival of palms after hurricane disturbances. Therefore, we added a new PFT for Palm. See also the response to Reviewer #1 Comment #1.

Comment #5 Line 69-71: we define a Palm PFT --> there is a need for a separate Palm PFT.

Re: See comment #4.

Comment #6 Line 85-86: maybe specify the version of ED2? ED-2.2 if citing Longo et al. 2019.

Re: The version used in this study is not exactly ED-2.2, it was downloaded from GitHub shortly before ED-2.2 was published. We cannot specify the exact version number of ED2 (a version between ED-2.1 and ED-2.2), but we specify the version number for the ED2-HuDi model: ED2-HuDi V1.0.

Comment #7 Line 159: Fig.2 uses time since disturbance to modify external seed rain rate (not seedling density). This assumes the recovery time scale is a constant. Why not use total LAI/BA? Early PFT seed rain can be high when LAI/BA is low but decreases when LAI/BA is high. This would be more ecologically meaningful.

Re: The recovery is not a constant as shown in Fig 2, it decreases with time since disturbance. Here, we intend to develop a seedling density that is explicitly affected by hurricane disturbances, i.e., changes with time since disturbance, and this time-varying behavior is also observed at our study site. LAI/BA could be a good indicator for seed rain or seedling density, but it is not an explicit indicator for hurricane disturbances.

Comment #8 Line 197: the definition of clumping factor is wrong. Should be effective LAI divided by total LAI.

Re: Revised.

Comment #9 Line 208-210 Any explanation for choosing stem density/DBH growth/BA as target state variables? Why not include mortality? There are large discrepancies between

simulated mortality (almost constant across years) and observed mortality (large inter-annual variability) in Fig. S5

Re: Because BA is directly calculated from the DBH of each cohort and weighted by the stem density of the cohort, the size structure (distribution of stem DBHs) of the forest is implicitly represented with the variables overall stem density and total BA. The PFT composition is explicitly represented with the PFT-specific variables. Therefore, stem density, BA, and DBH growth together describes the size structure and PFT composition of the forest (Lines 211-216). Mortality is not included because 1) it is implicitly included in the time series of the variable stem density; 2) it has higher uncertainty as typically few trees die each year, and this is more of an issue when calculating species-specific mortality (when the total number of trees is often very small), and a problem even at the PFT-level; and 3) mortality rates from observations are calculated from measurements that are years apart, which may cause the estimates to be lower than the true mortality because of missing records of mortality that trees were recruited after the previous survey and died before the next census. We agree that the discrepancies of the interannual variability of mortality between simulations and observations are large and we believe part of the reasons are that the observed mortality has uncertainties and are underestimated.

Comment #10 Line 328. Fig. 3. Are black dots observations or simulation? No information is provided in the caption. If they are observations like Fig. 4, what are the error bars, cross-plot variance? Also, I wonder how sensitive the results are to the length of training years. What about using half of the period as training?

Re: Black dots and error bars are the mean and standard deviation of observations across 85 plots, respectively, for both Figure 3 and Figure 4. Information has been added in the legend and caption. We tested different lengths of training years (# census years), including 10 years (3 census years 1989, 1994, and 1999), 15 years (4 census years between 1989 and 2004), 20 years (5 census years between 1989-2009) and 25 years (6 census years between 1989-2014; used for Fig. 3). The 20-year and 25-year training lengths give the same optimal result. The 10-year and 15-year training lengths have deteriorated performance on capturing the stem density of Early PFT (Figure AC3a and Figure AC4a). We will include the two figures in the supplementary information and add a brief description in the text.

> "We tested different calibration lengths (1989–1999, 1989–2004, and 1989-2009). 1989–2009 calibration period gives the same optimal simulation as 1989–2014 calibration period (Figure not shown), but shorter calibration lengths 1989–1999 (Figure AC3) and 1989–2004 (Figure AC4) throw away critical recovery information and cannot give robust simulation in the validation period."

[Figure]

Figure AC3: Same as Fig. 3, except that the optimal simulations are obtained by training 10 years (1989–1999) instead of 25 years (1989–2014).

[Figure]

Figure AC4: Same as Fig. 3, except that the optimal simulations are obtained by training 15 years (1989–2004) instead of 25 years (1989–2014).

Comment #11 Citation: Some of the most important information in methodology (such as allometric parameters, line 101) cite studies that are in review or in revision, and the paper only provides minimal information about them. There should be at least a brief description.

Re: Two of the three studies cited in review or in revision have been published now. The data for the H-DBH allometry are already published and accessible now, and we will show the allometric figure in the supplementary files, as in replies to Reviewer #1 Comment #15.

**References**

[revised manuscript text omitted]

---

## Author Response (AR2)

I appreciate the work done by the authors to address comments from myself and the other reviewers.

Re: Thanks!

**Anonymous Referee #3**

The authors have done a great job in addressing the comprehensive comments. The revised manuscript has improved readability and clarity. Overall, I think the study is providing useful information to model Palms and hurricane effects on vegetation dynamics.

A few further comments:

Comment #1: Fig. 3. Please clarify what does 'seedling density mean'? Does your model prescribe seedling density as a function of time or these are actually new recruitment due to internal/external seeds.

Re: The "seedling density" here means seedling density from seed rain, which are new recruitment due to external seeds that is a function of time since the last disturbance (Eq. (2)). We have revised the manuscript for clarity. Line 231.

> "**Figure 3.** The seedling density **from seed rain** for each PFT **as a function of time since disturbance**."

Comment #2: Fig. 6. For the GLUE results, what are the covariances between posterior parameters? I was wondering about the equifinality issue in such kind of inverse estimations, which might partly explain/interpret the high importance of light-related parameters?

Re: The figure below shows the correlation coefficients between any two posterior parameters. Instead of showing the covariances, we are showing the correlation coefficients, because the differences in variances among parameters are very large. For example, the variance is 19.64 $(\mu molCO2 \ m^{-2}s^{-1})^2$ for Early PFT Vm0 (carboxylation rate), but less than 0.0001 (unitless) for Rgf (growth respiration factor). Correlation coefficients are covariances normalized by the variances of the two variables involved and thus are more comparable among parameters. The figure shows that the posterior parameters are correlated (covariate) with each other, which is expected. Since this figure does not add more essential information to the manuscript, we will not include this figure in the revised manuscript.

[Figure]

Figure R1: The correlation coefficients between any two posterior parameters. The coefficients are calculated as the weighted correlation coefficients and the weights are those of each realization (calculated from Eq. (4)). The correlation coefficients that are significant at 99% level according to two-tailed *t*-test are shown with red (positive) and blue (negative).

We had discussed equifinality issue in Section 4.2, "*Given the nature of equifinality, there may be multiple parameter sets that can lead to the same observed state (Beven and Freer 2001), and thus the optimal parameter set we found from GLUE may be one of many possible solutions.*"

The light-related parameters are of high importance are largely because of the light limitation for plants in the model. Nutrients are assumed unlimited in the model, water is not limited in this tropical site, and light limitation matters the most, and thus light-related parameters contribute largely to model uncertainties. This is consistent with Meunier et al. (2021) and we have added a few sentences in the revised manuscript. Lines 608-612.

> "The optimization is sensitive to light-related parameters, such as clumping factor, quantum efficiency, and dark respiration (Figure 9). **This is possibly because light limitation is the most important limitation in the model, as water is not limited in this tropical site, and we turned off nutrient limitation.** This is consistent with Meunier et al. (2021) who found that light limitation contributes partly to model uncertainties."

Comment #3: Around Line 640 (ms with tracks) "Compared to other optimizers (such as PEcAn) that calibrates parameters using plant traits observations (e.g., wood density, leaf turnover rate), GLUE's ability of utilizing observations of forest stand variables (BA, AGB, etc.) could further reduce the uncertainty of parameters (Wang et al. 2013)" --> I am a little confused here. Doesn't PEcAn also have the ability to use BA/AGB to constrain the model in addition to plant traits observations? (e.g. Feng et al. 2018). Is this a true advantage of GLUE?.

Re: PEcAn was developed to "*synthesize plant trait data to estimate model parameters, propagate parameter uncertainties through to model output, and evaluate the contribution of each parameter to model uncertainty*" (LeBauer et al. 2013). The field measurements used in the PEcAn framework are those that are parameterized in the model (such as respiration rate, specific leave area, etc.) and parameter optimization/calibration is conducted before running model simulations. For BA and AGB, they are not parameters but model outputs, and thus they cannot be used for constraining model parameters in the PEcAn framework. In Feng et al. (2018), AGB was not used to constrain the model (or calibrate the parameters) but used to validate model simulations. We have revised the manuscript for clarity. Lines 600-603.

> "Compared to other optimizers (such as PEcAn) that calibrates parameters using plant traits observations (e.g., wood density, leaf turnover rate) **before running model simulations**, GLUE's ability of **constraining parameters from model output variables** utilizing observations of forest stand variables (BA, AGB, etc.) could further reduce the uncertainty of parameters (Wang et al. 2013)."

References

Beven, K. and Freer, J.: Equifinality, data assimilation, and uncertainty estimation in mechanistic modelling of complex environmental systems using the GLUE methodology, Journal of Hydrology, 249, 11–29, 2001.

Feng, X. et al.: Improving predictions of tropical forest response to climate change through integration of field studies and ecosystem modeling, Global Change Biology, 24, e213–e232, 2018.

LeBauer, D.S., Wang, D., Richter, K.T., Davidson, C.C., and Dietze, M.C.: Facilitating feedbacks between field measurements and ecosystem models, Ecological Monographs, 83, 133–154. 2013

Meunier, F. et al: Unraveling the relative role of light and water competition between lianas and trees in tropical forests: A vegetation model analysis, Journal of Ecology, 109, 519–540, 2021.

Wang, D., LeBauer, D. and Dietze, M.: Predicting yields of short-rotation hybrid poplar (Populus spp.) for the United States through model-data synthesis, Ecological Applications, 23, 944–958, 2013.

---

## Author Response (AR3)

**Editor**

I have checked your revisions and response letter. The manuscript is as good as ready to be published, I just want to see one additional correction:

In the response letter and in the revised manuscript you claim that PEcAn can only optimise parameters 'before simulations' (based on a meta-analysis of data on those parameters). This is not correct. I have double-checked this with people that have been using PEcAN intensively and PEcAn can be used both for (1) parameter optimisation 'before' simulations, and for (2) parameter optimisation by minimizing a cost function (minimizing the difference between observed and simulated output variables), similar as the GLUE approach. I am perfectly fine with the GLUE approach that has been used in your study. So please just adapt the statement about PEcAn (line 600-602) in that sense (or remove the comparison with PEcAn here).

Re: Dear Editor, thank you so much for the quality control and pointing out our mistake. We have now removed the comparison between GLUE and PEcAn (Lines 600-603):

> "Compared to other optimizers (such as PEcAn) that calibrates parameters using plant traits observations (e.g., wood density, leaf turnover rate) before running model simulations, GLUE's ability of constraining parameters from model output variables utilizing observations of forest stand variables (BA, AGB, etc.) could further reduce the uncertainty of parameters (Wang et al. 2013)."

and corrected a citation (Line 218):

> "Everham **and Brokaw** 1996"